# *Ehrlichia* SLiM ligand mimetic activates Hedgehog signaling to engage a BCL-2 anti-apoptotic cellular program

**Caitlan D. Byerly**[1], **Shubhajit Mitra**[1], **LaNisha L. Patterson**[1], **Nicholas A. Pittner**[1], **Thangam S. Velayutham**[1], **Slobodan Paessler**[1,2], **Veljko Veljkovic**[2], **Jere W. McBride**[1,3,4,5,6]*

1 Department of Pathology, University of Texas Medical Branch, Galveston, Texas, United States of America, 2 Biomed Protection, LLC, Galveston, Texas, United States of America, 3 Department of Microbiology and Immunology, University of Texas Medical Branch, Galveston, Texas, United States of America, 4 Center for Biodefense and Emerging Infectious Diseases, University of Texas Medical Branch, Galveston, Texas, United States of America, 5 Sealy Institute for Vaccine Sciences, University of Texas Medical Branch, Galveston, Texas, United States of America, 6 Institute for Human Infections and Immunity, University of Texas Medical Branch, Galveston, Texas, United States of America

* jemcbrid@utmb.edu

**Data Availability Statement:** All relevant data are within the manuscript.

**Funding:** This work was supported by the National Institute of Allergy and Infectious Disease (https://

## Abstract

*Ehrlichia chaffeensis* (*E. chaffeensis*) has evolved eukaryotic ligand mimicry to repurpose multiple cellular signaling pathways for immune evasion. In this investigation, we demonstrate that TRP120 has a novel repetitive short linear motif (SLiM) that activates the evolutionarily conserved Hedgehog (Hh) signaling pathway to inhibit apoptosis. *In silico* analysis revealed that TRP120 has sequence and functional similarity with Hh ligands and a candidate Hh ligand SLiM was identified. siRNA knockdown of Hh signaling and transcriptional components significantly reduced infection. Co-immunoprecipitation and surface plasmon resonance demonstrated that rTRP120-TR interacted directly with Hh receptor Patched-2 (PTCH2). *E. chaffeensis* infection resulted in early upregulation of Hh transcription factor GLI-1 and regulation of Hh target genes. Moreover, soluble recombinant TRP120 (rTRP120) activated Hh and induced gene expression consistent with the eukaryotic Hh ligand. The TRP120-Hh-SLiM (NPEVLIKD) induced nuclear translocation of GLI-1 in THP-1 cells and primary human monocytes and induced a rapid and expansive activation of Hh pathway target genes. Furthermore, Hh activation was blocked by an α-TRP120-Hh-SLiM antibody. TRP120-Hh-SLiM significantly increased levels of Hh target, anti-apoptotic protein B-cell lymphoma 2 (BCL-2), and siRNA knockdown of BCL-2 dramatically inhibited infection. Blocking Hh signaling with the inhibitor Vismodegib, induced a pro-apoptotic cellular program defined by decreased mitochondria membrane potential, significant reductions in BCL-2, activation of caspase 3 and 9, and increased apoptotic cells. This study reveals a novel *E. chaffeensis* SLiM ligand mimetic that activates Hh signaling to maintain *E. chaffeensis* infection by engaging a BCL-2 anti-apoptotic cellular program.

www.niaid.nih.gov/) grants AI137779 and AI149136 to J.W.M., McLaughlin Endowment and T32AI007526-20 Biodefense Training Program (https://www.utmb.edu/mclaughlin/) predoctoral fellowships to C.D.B., NIH 1F31AI152424 predoctoral fellowship (https://www.ninds.nih.gov/Funding/Training-Career-Development/Award/F31-Individual-NRSA-PhD-Students-MDPhD-Students-MSTP-0) to L.L.P., and Sealy Center for Vector Borne and Zoonotic Diseases (https://www.utmb.edu/pathology/research/the-center-for-vector-borne-and-zoonotic-diseases/grad-student-fellowships) predoctoral fellowship to N.A.P. The funders had no role in study design, data collection and analysis, decision to publish, or preparation of the manuscript.

**Competing interests:** The authors have declared that no competing interests exist.

## Author summary

*Ehrlichia chaffeensis* is an obligately intracellular bacterium that preferentially infects and replicates within mononuclear phagocytes and survives intracellularly by modulating cellular signaling pathways to subvert innate immune defenses. This investigation reveals the complex and expanding role that the *E. chaffeensis* TRP120 moonlighting effector and SLiM ligand mimetics have on immune subversion and infection through activation and regulation of evolutionarily conserved signaling pathways. Herein, we define a TRP120-Hh-SLiM mimetic that induces Hh signaling and regulates the anti-apoptotic protein BCL-2 to prevent sequential activation of caspase 9 and 3, promoting *E. chaffeensis* infection. This study defines a novel prokaryotic SLiM mimetic that repurposes evolutionarily conserved eukaryotic signaling pathways to promote survival of an intracellular bacterium.

## Introduction

*Ehrlichia chaffeensis* (*E. chaffeensis*) is an obligately intracellular tick-borne rickettsial pathogen and the etiologic agent of human monocytotropic ehrlichiosis (HME), an emerging, life-threatening zoonosis. Mononuclear phagocytes are preferentially infected by *E. chaffeensis*, where it replicates in early endosome-like host membrane-derived cytoplasmic vacuoles and completes a biphasic intracellular life cycle [1]. During infection, *E. chaffeensis* secretes well characterized tandem repeat protein (TRP) effectors via the type 1 secretion system (T1SS) which interact with a diverse array of host proteins [2]. TRP-host interactions rewire fundamental host cell processes such as gene transcription and activation of conserved cellular signaling pathways (Wnt and Notch), through interactions that involve post-translational modifications and SLiM mimicry to reprogram the host cell and subvert innate immune defenses [3–10].

*Ehrlichia chaffeensis* TRP120 has been recognized as a moonlighting protein that has multiple roles during infection [11]. Initially, TRP120 was identified as a major immunoreactive protein, but was later detected in the nucleus where it functions as a nucleomodulin. Subsequently, yeast-2-hybrid (Y2H) analysis identified a multitude of molecular interactions between TRP120 and eukaryotic proteins involved in various cellular processes, including cell signaling, transcriptional regulation, PTMs and apoptosis [10]. Recently, multiple TRP120 functions have been well defined including DNA binding [12], HECT E3 ubiquitin ligase activity [4,13], and ligand mimicry [3]. Notably, TRP120 engages Notch and Wnt receptors to activate conserved Notch and Wnt signaling through SLiM ligand mimicry [3,5]. SLiM ligand mimicry and post translational modification (PTM) SLiMs have been well characterized in bacteria and viruses [3,14,15]. However, examples of interkingdom Hedgehog (Hh) SLiM mimicry have never been reported.

The Hh pathway is an evolutionarily conserved signaling pathway that plays a crucial role in embryogenesis [16]. The pathway was first identified in *Drosophila* and extensively studied in the field of developmental biology for its role in segment polarity and body patterning during embryogenesis [17]. Hh signaling components are highly conserved in vertebrates and invertebrates, but the key difference is in pathway redundancy [18]. In *Drosophila* there is one ligand (Hh), one primary receptor (PTC) and one transcription factor (Ci). However, in mammals the Hh pathway has three families of ligands: Sonic hedgehog (Shh); Indian hedgehog (Ihh) and Desert hedgehog (Dhh) two primary receptors (PTCH1 and PTCH2) and three transcription factors (GLI-1, GLI-2, and GLI-3) [19,20]. Hh signaling is initiated by the Hh ligand

binding to the Hh receptor Patched (PTCH). This interaction counters PTCH-mediated repression of Smoothened (SMO) and activates the only known transcriptional mediators, the GLI family of Hh transcription factors [21]. Regardless, the signal is relayed from external milieu to nucleus in a moderately conserved way [17]. In humans, the Hh pathway is involved in maintaining tissue homeostasis, and aberrant activation of this pathway results in the formation of various tumors and hematological malignancies [22,23]. Hh has also been identified as a key regulator in maintaining tissue homeostasis and remodeling due to various roles associated with cell proliferation, angiogenesis, B and T cell development, regulation of immune response, autophagy and cellular apoptosis [24]. Notably, evolutionary conserved signaling pathways such as Hh are known to be engaged by pathogens. Viral and bacterial pathogens including Hepatitis B and C (HBV and HCV, respectively), Epstein–Barr virus (EBV), Influenza A virus, *Helicobacter pylori*, and *Mycobacterium bovis* have recently been reported to modulate Hh signaling during infection [20]. As cellular apoptosis is highly regulated by Hh [25], exploiting the Hh pathway may be an important strategy for intracellular pathogens to enhance host cell survival to promote intracellular infection [26].

Apoptosis is the default programmed cell death mode for organ development during embryogenesis and has emerged as one of the major pathways controlled by Hh signaling [24,27]. Moreover, apoptosis also plays an important role in host immune defense during microbial infections, triggering the sequential activation of caspases, in response to either an extrinsic or intrinsic death signal [28]. The release of cytochrome c from the mitochondria initiates the intrinsic apoptotic pathway and results in cytochrome c association with apoptotic protease activating factor 1 (Apaf-1) and pro caspase 9 to form the apoptosome, a multimeric protein complex involved in cleavage of inactive caspase 9 into the active form [29]. Caspase 9 cleaves pro caspase 3, to form active caspase 3, which is essential for chromatin condensation and DNA fragmentation in all apoptotic cells [30]. One of the major anti-apoptotic transcriptional targets of GLI-1 is BCL-2, which is essential in inhibiting apoptosis inducer Bax [31,32]. The anti-apoptotic members of the BCL-2 family maintain mitochondrial membrane integrity by sequestering BH3-only proteins like Bax and Bak to inhibit the release of cytochrome c from mitochondria, thereby inhibiting apoptosis [33]. Moreover, *E. chaffeensis* influences the transcriptional activity of other Hh-targeted anti-apoptotic genes such as *BCL-2A*, *MCL1*, and *BIRC3* [4,34,35].

This investigation reveals a novel host-pathogen strategy, whereby *E. chaffeensis* utilizes eukaryotic SLiM mimicry to exploit Hh signaling to activate an anti-apoptotic cellular program. We determined that *E. chaffeensis* TRP120 directly interacts with Hh receptor, PTCH2 and Hh signaling is activated through a Hh ligand SLiM mimetic. Furthermore, we analyzed the role of Hh signaling in regulating apoptosis during infection and demonstrate that *E. chaffeensis* exploits Hh signaling to engage BCL-2 and inhibit apoptosis during infection.

## Results

### *E. chaffeensis* TRP120 contains a predicted Hh ligand SLiM

TRP120 contains a tandem repeat domain (TRD) centered between the N- and C-terminal domains. Various functional SLiMs have been reported within the N terminus, TRD, and C terminus that are relevant to *E. chaffeensis* infection, including posttranslational modification motifs, DNA-binding motifs, and ubiquitin ligase catalytic motifs [3]. We previously reported that TRP120 stimulation results in transcriptional upregulation of *GLI-1* in THP-1 cells [36]. Moreover, during *E. chaffeensis* infection Wnt and Notch signaling pathways are activated through ligand mimicry via TRP120 Wnt and Notch SLiMs [3,5]. Since GLI-1 is a major transcriptional factor of the Hh signaling pathway and often cross-talks with Wnt and Notch

signaling [37,38], we investigated the possibility of Hh pathway activation during *E. chaffeensis* infection. Using NCBI Protein BLAST and informational spectrum method (ISM) analysis, we determined that TRP120 has sequence homology and predicted functional similarity with Hh ligands respectively within the TRD (**Fig 1**). Specifically, the homologous TRP120 Hh sequence (NPEVLIKD) present in each TR is 87% similar to the Hh ligand sequence that is associated with the Hh-PTCH binding site [39]. In addition, the homologous TRP120 Hh sequence is 8 aa, consistent in length with other known SLiMs (**Fig 1A**). Therefore, the homologous TRP120 Hh sequence was designated as the TRP120-Hh-SLiM.

While sequence homology can be a useful tool to determine whether a bacterial protein may mimic a eukaryotic motif, amino acid sequence homology does not suggest functional similarity. To predict functional similarity between TRP120 and Hh ligands, we performed ISM analysis comparing TRP120 to Dhh and Ihh (**Fig 1B–1D**). ISM analysis is performed *in silico* to identify shared characteristics between two molecules by predicting similar long-wave frequency vibrations that dictate various protein functions [40]. TRP120-Dhh (**Fig 1B**) and TRP120-Ihh (**Fig 1C**) ISM cross-spectrum electron-ion interaction potentials (EEIP) detected significant peak amplitude at frequency 0.457, predicting a shared biological function between TRP120 with Dhh and Ihh in the same region. Scanning the EEIP sequence of TRP120 along the peak amplitude frequency identified the amino acids shared between TRP120 with Dhh and Ihh. The amino acids reside within the TRD of TRP120, immediately upstream of the

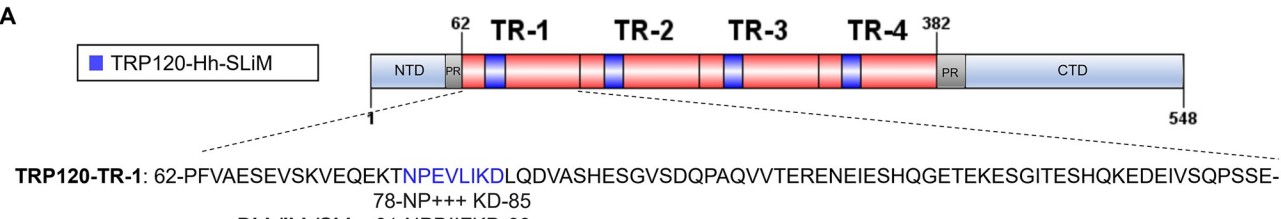

**TRP120-TR-1**: 62-PFVAESEVSKVEQEKTNPEVLIKDLQDVASHESGVSDQPAQVVTERENEIESHQGETEKESGITESHQKEDEIVSQPSSE-141
 78-NP+++ KD-85
**Dhh/Ihh/Shh**: 81-NPDIIFKD-88

| Dhh/Ihh/Shh Sequence | TRP120-Hh-SLiM Sequence | Similarity | Repeats |
|---|---|---|---|
| NPDIIFKD | NPEVLIKD | 87% | 4 |

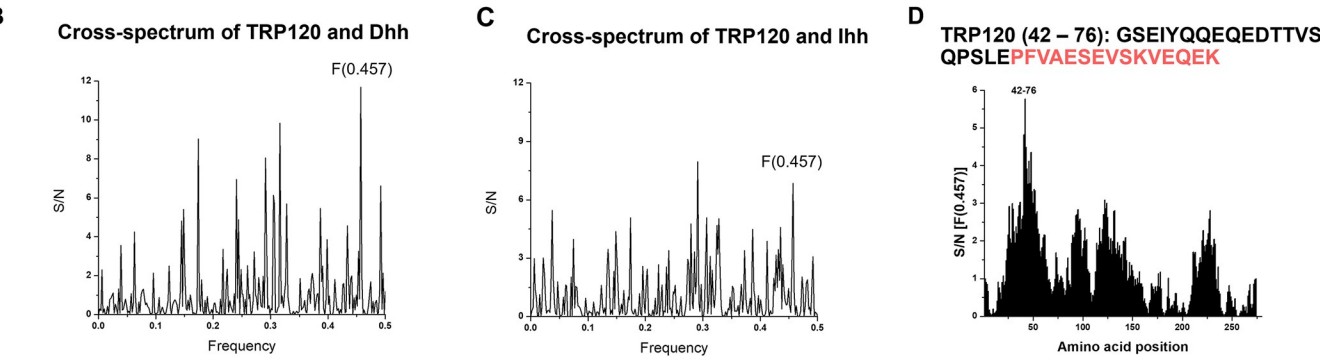

**Fig 1. TRP120 TR motif shares sequence and functional similarity with Hh ligands.** (A) Schematic representation of TRP120 showing domain organization. TRP120 consists of a N-terminal (NTD), C-terminal (CTD) and tandem repeat (TR1–4; 80 aa each) domain, flanked with two partial repeats (PR) [89]. NCBI Protein BLAST identified an 8 amino acid short linear motif (SLiM) of high similarity between the TRP120 TR and Hh ligand amino acid sequences corresponding to the location of Hh ligand and Hh PTCH receptor-binding site [39,63]. Complete amino acid sequence of one TR is shown with homologous Hh SLiM identified in blue. Blue shaded TR regions shown in schematic indicate location of Hh SLiM. Table summarizes amino acid sequence similarity between Hh ligands and TRP120; (B-D) ISM analysis of TRP120 and Dhh/Ihh; (B) Cross-spectrum of TRP120 (accession no. AAO12927.1) and Dhh (NP_066382) with ISM frequencies (x axis) plotted against normalized amplitude for each component (y axis); (C) Cross-spectrum of TRP120 and Ihh (NP_002172) with ISM frequencies (x axis) plotted against normalized amplitude for each component (y axis); (D) Scanning of the amino acid sequence of Dhh/Ihh along ISM F[0.457] immediately upstream of the TRP120-Hh-SLiM. The TRP120 TR region is defined in red.

predicted TRP120-Hh-SLiM (**Fig 1D**). Together, the BLAST and ISM analysis results suggest that TRP120 contains a Hh SLiM within the TRD that has sequence similarity with the receptor-binding site of Hh ligands and is predicted to have functional similarity with Hh ligands by ISM.

## Hh signaling components are required for *E. chaffeensis* survival

The Hh signaling pathway is not only required during embryogenesis, but also plays a major role in determining cell fate in adult hematopoietic cells. Since Hh signaling is involved in different cellular processes like autophagy and apoptosis [41,42], which are crucial for ehrlichial intracellular survival [43], we examined the effect of Hh signaling inhibition on *E. chaffeensis* infection using iRNA to individually target and silence *GLI-1/2/3*, *PTCH1*, *PTCH2* and *SMO* in THP-1 cells. *E. chaffeensis* infection, determined by by ehrlichial disulfide bond formation protein (*dsb)* gene copies, was significantly reduced in nearly all transfection groups 24 h post transfection of siRNA (excluding *PTCH1*-knockdown cells), relative to the infection level in scrambled siRNA-transfected cells (**Fig 2A**). The most significant impact on *E. chaffeensis* infection occurred in *GLI-1-*, *PTCH2-* and *SMO*-knockdown cells. Loss of PTCH2 receptor significantly reduced infection, while loss of PTCH1 receptor did not, suggesting that *E. chaffeensis* may preferentially target PTCH2 during infection.

## TRP120 interacts with Hh receptor PTCH2

Hh signaling initiates when a Hh ligand binds to the PTCH receptor, disengaging PTCH-mediated inhibition of SMO, which results in nuclear translocation of the full-length GLI-1 transcription factor and subsequent activation of Hh pathway target genes [18]. A Hh SLiM mimetic was identified and iRNA knockdown studies performed suggests that *E. chaffeensis* favors PTCH2 for Hh activation. Based on these results, we hypothesized that TRP120 is a Hh ligand mimic that directly interacts with PTCH2. To examine the cellular distribution and colocalization of PTCH2 with the TRP120-expressing ehrlichial inclusions, cells were stained

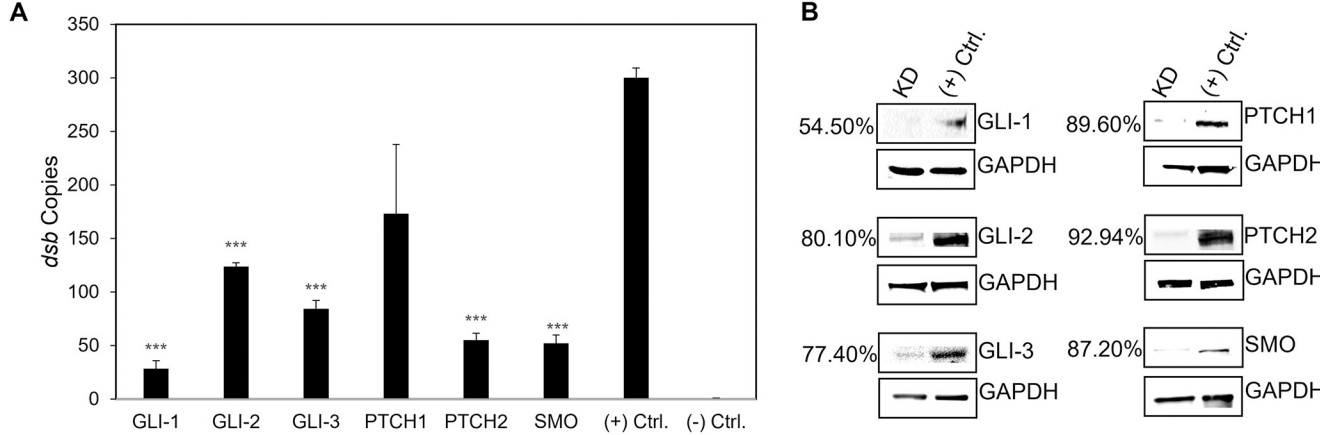

**Fig 2. iRNA knockdown of Hh signaling components inhibits *E. chaffeensis* infection.** (A) Small interfering RNA-transfected (siRNA) THP-1 cells were infected or mock infected (-) with *E. chaffeensis* (MOI 100, 24 h post-transfection). Scrambled siRNA (scrRNA) was transfected for positive (infected) and negative control (mock infected*). E. chaffeensis* infection was quantified at 24 hpi and was determined by qPCR amplification of the *dsb* gene. siRNA knockdown (KD) of Hh receptors PTCH2, SMO, and transcription factors GLI-1/2/3 significantly inhibits *E. chaffeensis* infection in THP-1 cells. All knockdowns were performed with at least three biological and technical replicates for *t*-test analysis. Data are represented as means ± SD (***p<0.001). (B) Western blot depicts knockdown efficiency of siRNA in knockdown cells compared to positive control from whole-cell lysates harvested 24 hpi. Number left of siRNA lane indicates percent knockdown of protein of interest relative to positive control, normalized to GAPDH expression.

with anti-PTCH2 and anti-TRP120 specific antibody and observed by immunofluorescence microscopy. We found a mostly punctate distribution of PTCH2 receptors in uninfected THP-1 cells; however, in infected cells, we found colocalization of PTCH2 with morulae expressing TRP120 (**Fig 3A**). Intensity correlation analysis using ImageJ demonstrated a positive Pierson's correlation coefficient (PCC = 0.866) between PTCH2 and TRP120. In addition, colocalization of native PTCH2 receptor and ectopically expressed GFP-TRP120 in transfected HeLa cells supports that an interaction exists between TRP120 and PTCH2 (**Fig 3B**). Since these data only indicate TRP120 colocalization with PTCH2, we performed two protein-interaction assays, including Co-Immunoprecipitation (Co-IP) and surface plasmon resonance (SPR). We confirmed the direct interaction between TRP120 and PTCH2 by immunoprecipitating

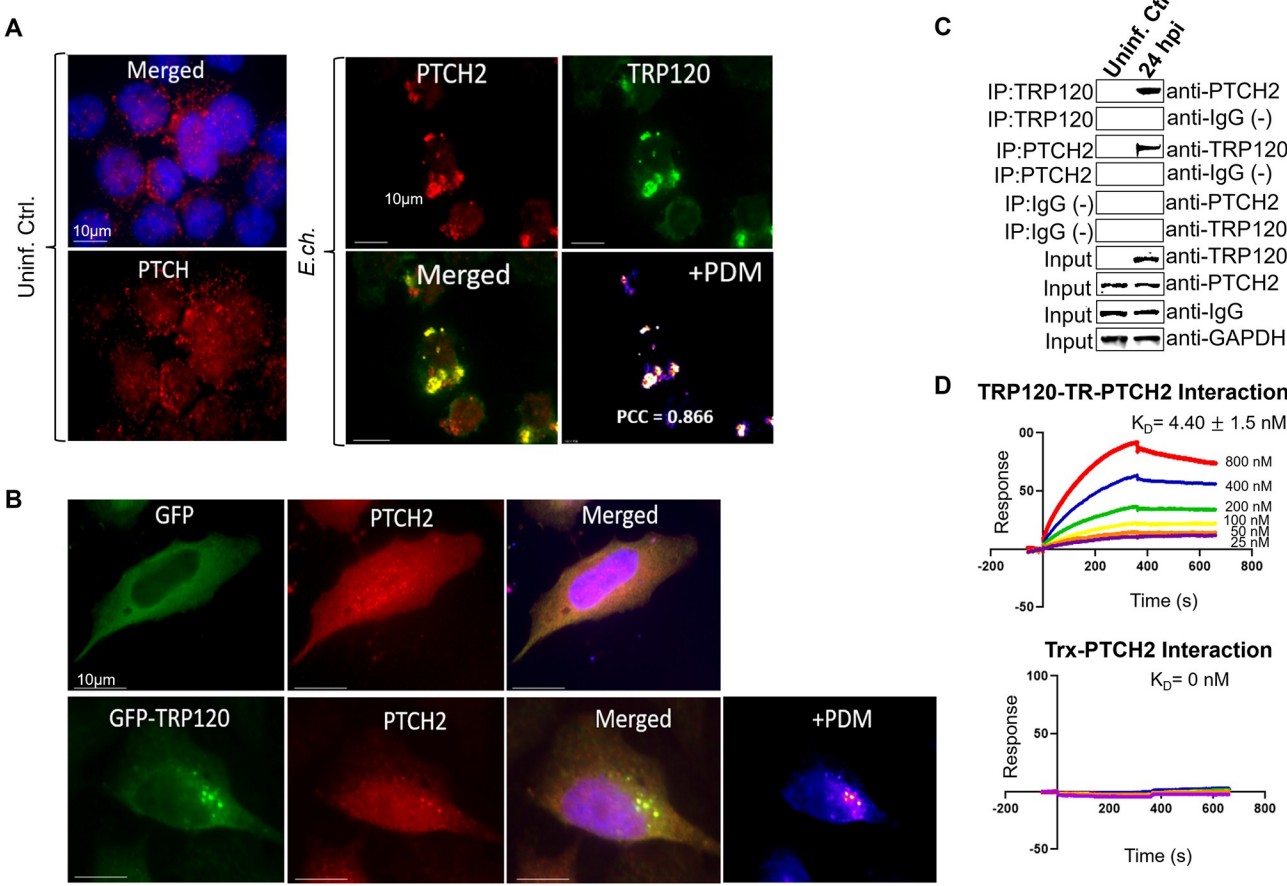

**Fig 3. TRP120 interacts directly with Hh receptor PTCH2.** (A) Immunofluorescence micrographs showing the distribution of TRP120 and PTCH2 receptor in uninfected and *E. chaffeensis*-infected cells (MOI 100). Co-localization of *E. chaffeensis* expressing TRP120 (green) with Hh receptor PTCH2 (red) at the ehrlichial inclusion was observed at 48 hpi compared to uninfected cells. Pearson's correlation coefficient (PCC) indicates a correlation between TRP120 and PTCH2, suggesting a direct interaction. Product of the difference from Mean (PDM) displays an average positive pixel intensity (scale bar, 10μm). (B) Immunofluorescence analysis of GFP tagged TRP120 transfected HeLa cells. GFP-TRP120 or GFP (ctrl) expressing HeLa cells were immunostained with PTCH2 specific antibody and observed by epifluorescence microscopy. Colocalization of GFP-tagged TRP120 (green) with the PTCH2 receptor (red) was observed in HeLa cells. (A-B) Experiments were performed with at least three biological and technical replicates. Randomized areas/slide (n = 10) were used to detect interaction. (C) Co-IP and reverse Co-IP demonstrate the direct interaction between TRP120 and PTCH2 at 24 hpi compared to the IgG negative control. Western blot analysis was normalized to GAPDH expression and experiment was repeated with three biological replicates. (D) Surface plasmon resonance (SPR) was utilized to detect the direct interaction of rPTCH2 with rTRP120-TR or rPTCH2 with rTrx (-). rPTCH2-His was immobilized on a nickel chip followed by the injection of an analyte solution containing solubilized rTRP120-TR or rTrx (-) at 2-fold dilutions (800nM to 25nM) to determine the binding affinity ($K_D$). The binding affinity demonstrates an interaction between rPTCH2 and rTRP120-TR. The results were expressed as the means ± standard deviation (SD) of data obtained from three independent experiments.

TRP120 or PTCH2 (reverse Co-IP) from the lysate of infected THP-1 cells harvested at 0 hpi (uninfected control) and 24 hpi, which ensured that sufficient levels of TRP120 were present (**Fig 3C**). An interaction between PTCH2 and TRP120 was detected with Co-IP which was also demonstrated with reverse Co-IP. Additionally, SPR was utilized to confirm a direct interaction between TRP120 TRD and PTCH2 and determine the binding affinity (**Fig 3D**). An interaction between rPTCH2 and rTRP120-TR ($K_D = 4.40 \pm 1.5$ nM) was detected compared to the negative control ($K_D = 0$). These data demonstrate that TRP120 directly interacts with PTCH2 via the TRD (**Fig 3**), which contains both sequence and functional similarity with Hh ligands (**Fig 1**). These results identified PTCH2 as a receptor for TRP120.

## *E. chaffeensis* activates the Hh signaling pathway in THP-1 cells and primary human monocytes

Hh signal initiates at the plasma membrane when Hh ligands interact with the 12-pass-transmembrane PTCH receptor. The Hh ligand-PTCH interaction results in increased expression of cell surface receptor SMO, decreased levels in the cytoplasmic negative regulator SUFU, and subsequent activation and nuclear translocation of Hh transcription factor, GLI-1 [44–46]. We predicted an *E. chaffeensis* TRP120-Hh-SLiM and identified a direct interaction between TRP120 and PTCH2 during infection. Additionally, we previously reported that TRP120 stimulation results in transcriptional upregulation of *GLI-1* in THP-1 cells [36]. However, the role of *E. chaffeensis* in activating the Hh signaling pathway has not been defined. Hence, we investigated whether *E. chaffeensis* upregulates GLI-1 in THP-1 cells and primary human monocytes via confocal microscopy (**Fig 4**). We first determined that *E. chaffeensis* stimulates GLI-1 upregulation and nuclear translocation in THP-1 cells. GLI-1 was detected in the nucleus within 2 hpi, and progressive nuclear accumulation of GLI-1 was observed over 48 hpi compared to uninfected controls at respective timepoints (**Fig 4A and 4C**). Further, *E. chaffeensis*-infected primary human monocytes stimulated GLI-1 upregulation and nuclear translocation at 10 hpi compared to the uninfected control, which provided further evidence that *E. chaffeensis* activates Hh signaling (**Fig 4B and 4D**).

## Expression array analysis of Hh-signaling genes during *E. chaffeensis* infection

To further examine the role of *E. chaffeensis* in Hh pathway activation, we examined Hh pathway gene transcription during *E. chaffeensis* infection. A transcriptional analysis was performed using a human Hh signaling PCR array, including Hh components, putative targets, and auxiliary genes at 4, 8, 24 and 48 hpi (**Fig 5**). Volcano plots generated from data sets at 4, 8, 24, and 48 hpi depict a differential expression pattern of Hh signaling pathway genes in the *E. chaffeensis*-infected cells compared to uninfected cells (**Fig 5A**). Significant activation of Hh pathway regulator, component, and target genes was detected between 4, 8, 24, and 48 hpi. Only a small number of Hh-associated genes were negatively regulated during infection. The expression patterns of genes that showed a consistent and a significant upregulation throughout all different time points included Hh pathway regulators: *BOC, CDON, BTRC, CSNK1E* and *PRKACA;* Hh signaling pathway auxiliary genes: *LATS1, MAPK1* and *NF2* and the Hh pathway target genes: *MTSS1, WNT10A, WNT3, WNT9a and VEGFA*. The core Hh signaling pathway receptor genes like *PTCH1, PTCH2 and SMO* were highly expressed during early and late time points, suggesting high pathway activity throughout *E. chaffeensis* infection. One of the major anti-apoptotic genes and a major target of Shh-signaling pathway *BCL-2* showed transcriptional upregulation at 24 and 48 hpi suggesting a crucial role of Hh signaling pathway in inhibition of host cell apoptosis during *E. chaffeensis* infection. Normalized expression of

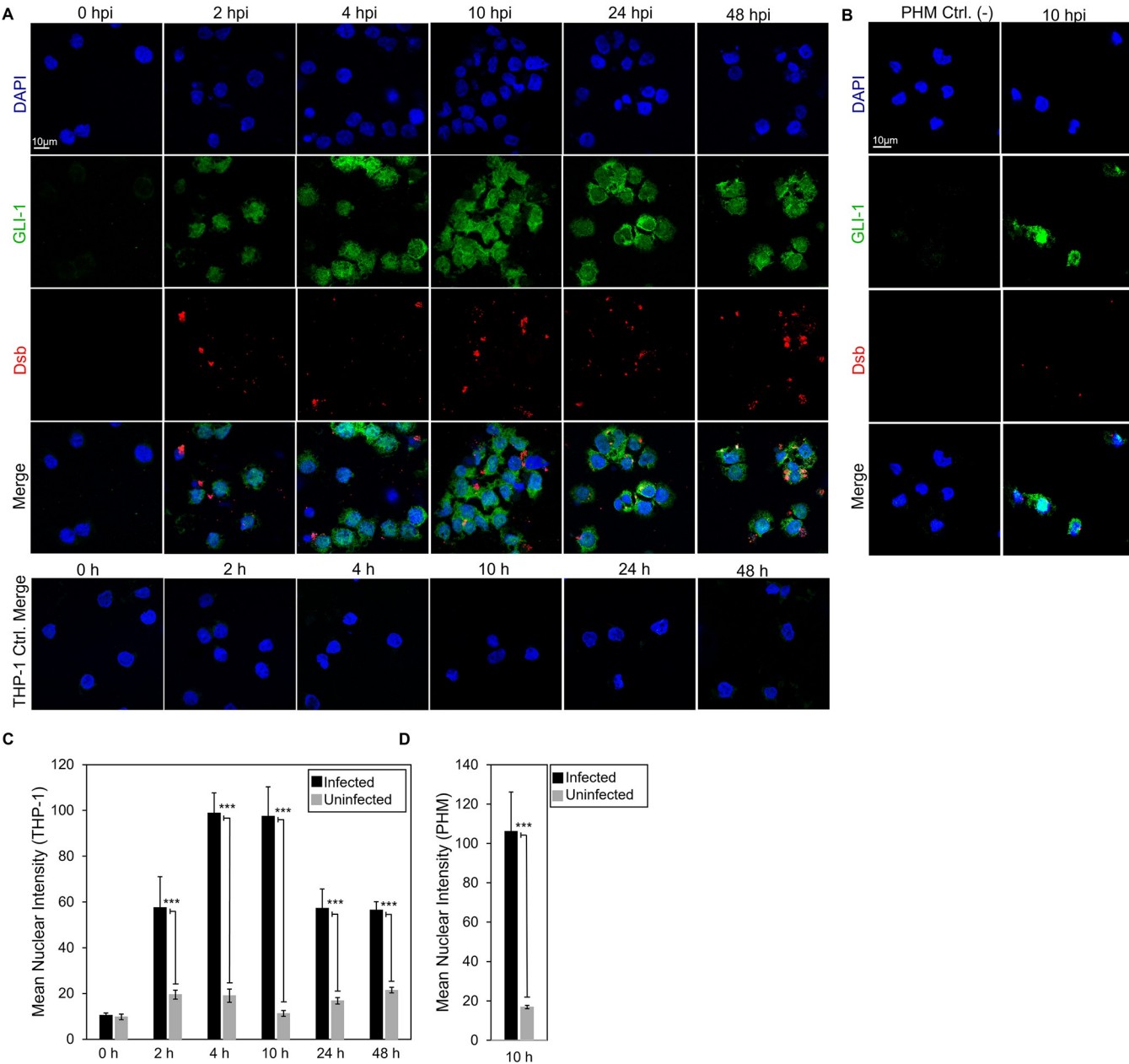

**Fig 4. *E. chaffeensis* activates the Hh signaling pathway in THP-1 cells and primary human monocytes.** (A) Confocal microscopy of uninfected and *E. chaffeensis* infected THP-1 cells stained with anti-GLI-1 antibody at 2, 4, 10, 24 and 48 hpi, showing a temporal increase in GLI-1 compared to the uninfected controls. Uninfected and *E. chaffeensis* infected (MOI 100) THP-1 cells were stained with anti-Dsb antibody (red) at 2, 4, 10, 24 and 48 hpi demonstrating infection of THP-1 cells with *E. chaffeensis* (scale bar = 10 μm). (B) Confocal microscopy of uninfected and *E. chaffeensis*-infected primary human monocytes at 10 hpi showing upregulation of GLI-1 (green). Uninfected and *E. chaffeensis*-infected primary human monocytes were stained with anti-Dsb antibody (red) to confirm *E. chaffeensis* infection (scale bar = 10 μm). (A-B) Experiments were performed with at least three biological and technical replicates. Randomized areas/slide (n = 10) were used to detect GLI-1 nuclear translocation. (C-D) Intensity graphs demonstrate the mean nuclear accumulation of GLI-1 in respective THP-1 cells and primary human monocytes. Analysis was performed using ImageJ and determining mean grey value from randomized areas/slide (n = 10).

selected genes in the Hh PCR array between infected and uninfected cells at 48 hpi is shown in **Fig 5B**. We also confirmed immunofluorescent microscopy results by probing nuclear fractions of uninfected and *E. chaffeensis*-infected THP-1 cells with GLI-1 specific antibody by immunoblot. Progressive nuclear accumulation of GLI-1 was observed over 48 hpi

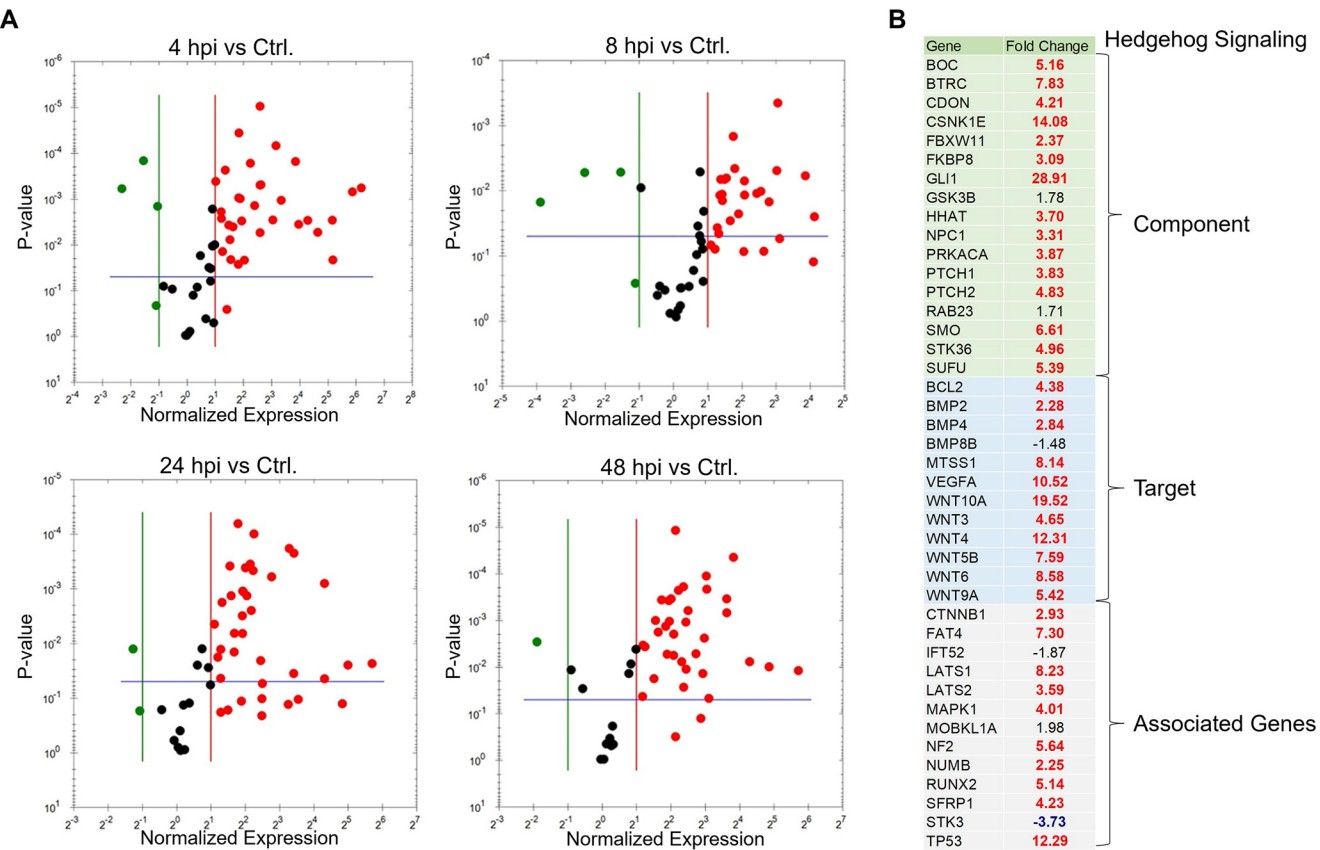

**Fig 5. Expression array analysis of Hh-signaling genes during *E. chaffeensis* infection.** (A) Volcano plots showing the differential expression of Hh-signaling pathway genes between *E. chaffeensis*-infected (MOI 100) and uninfected THP-1 cells at 4 h (top left), 8 h (top right), 24 h (bottom left) and 48 hpi (bottom right). The red, black and green dots in the volcano plot represent upregulation ($\geq 2$), no change, and down-regulation ($\leq$ -2), respectively. The horizontal blue lines on the volcano plots determine the level of significance ($\leq 0.05$). Genes were significantly upregulated at all time points. (B) Normalized expression of Hh array genes in component, target and associated gene categories between *E. chaffeensis*-infected and uninfected cells at 48 hpi. Cells were harvested with three biological and technical replicates for all time points.

(**Fig 6A and 6B**). In addition, we tested cytoplasmic fractions of SMO during *E. chaffeensis*-infection and found induced SMO protein expression during infection. In addition, we also detected decreased protein levels of cytoplasmic GLI-1 negative regulator SUFU and increased protein levels of Shh in *E. chaffeensis*-infected THP-1 cytoplasmic fractions (**Fig 6A and 6B**). Collectively, these data demonstrates that *E. chaffeensis* activates the Hh-signaling pathway during infection.

## TRP120 upregulates GLI-1 and Hh gene expression consistent with Hh ligands

To further examine the role of TRP120 in Hh pathway activation, purified rTRP120-FL (1 μg/ml) was used to stimulate THP-1 cells and primary human monocytes, and cellular expression and distribution of GLI-1 were monitored using confocal microscopy (**Fig 7A and 7B**). GLI-1 upregulation, accumulation and nuclear translocation was observed in THP-1 cells (**Fig 6A and 6C**) and primary human monocytes (**Fig 7B and 7D**) at 6 and 10 hpt, respectively. THP-1 cells and primary human monocytes treated with rTRP120-FL demonstrated clear GLI-1 upregulation, similarly to recombinant Shh (rShh), which was used as a positive control. To further

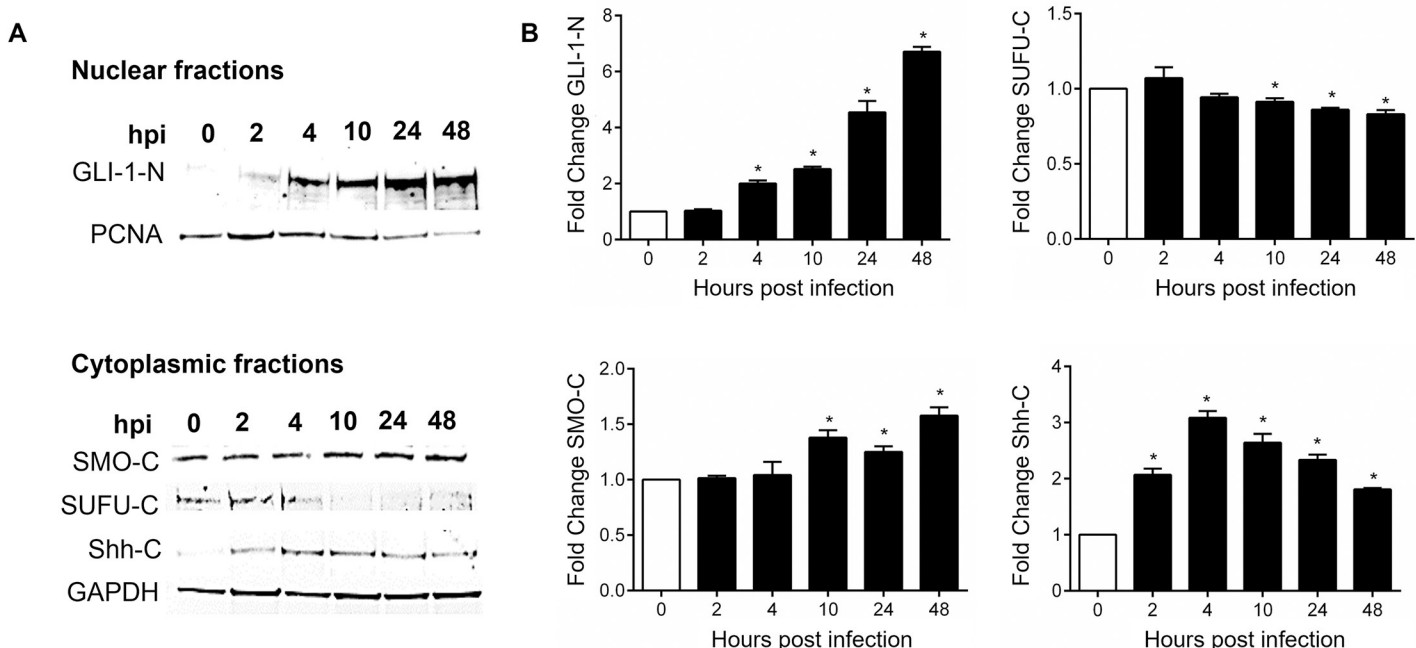

**Fig 6. *E. chaffeensis* infection alters levels of Hh-signaling components in THP-1 cells.** (A) Western blot analysis of GLI-1 levels in uninfected and *E. chaffeensis*-infected THP-1 cell nuclear fractions (N) collected at 0, 2, 4, 10, 24 and 48 hpi with PCNA as a nuclear control. Western blot analysis of cytoplasmic fraction levels (C) of SMO, SUFU and Shh in uninfected and *E. chaffeensis*-infected THP-1 cells at 0, 2, 4, 10, 24 and 48 hpi with GAPDH as a loading control. (B) Bar graphs depicting Western blot densitometry values were normalized to PCNA or GAPDH, respectively. (A-B) Western blots were performed with at least three biological and technical replicates for *t*-test analysis. Data are represented as means ± SD (\*$p < 0.05$).

confirm the role of TRP120 in Hh pathway activation, cells were stimulated with rTRP120-FL or rShh for 24 h, and transcriptional analysis was performed using a human Hh signaling PCR array (**Fig 7E and 7F**). The volcano plots represent gene expression patterns of all 84 genes in the Hh signaling PCR array in cells stimulated with rTRP120-FL normalized to negative control cells treated with rTrx. A significant increase in 15 Hh genes, including Hh pathway associated receptors, and cofactors (*PTCH2*, *SMO*, *CDON* and *LRP2*), regulators (*BTRC*, *CSNK1E* and *PRKACA*), transcription factor (*GLI1*), and target genes (*WNT10A*, *WNT6*, *WNT2B* and *WNT4*) was detected (**Fig 7E**). In addition, 2 genes (*WIF1* and *GAS1*) were downregulated compared to the control (**Fig 7E**). In comparison, cells treated with rShh had a significant increase in 17 genes in the Hh pathway array (**Fig 7F**). Though there were differential expression pattern of genes in TRP120 and Shh treated cells, we found 8 Hh pathway-associated genes including (*PTCH2*, *SMO*, *GLI1*, *CSNK1E, and LRP2*) and target genes (*WNT10A*, *WNT4, and WNT6*) were upregulated in both rTRP120-FL and rShh treatment. Together these data demonstrate that TRP120 independently and efficiently activates the Hh signaling pathway.

## TRP120-Hh-SLiM activates Hh signaling in THP-1 cells and primary human monocytes

We next investigated if the predicted TRP120-Hh-SLiM sequence (NPEVLIKD) was sufficient in activating Hh signaling (**Fig 8**). A table was used to reveal various TPR120 peptide sequences within the TRP120 TRD domain. TRP120-TR-Hh (20 aa) and TRP120-Hh-SLiM (8 aa) sequences contain the TRP120-Hh homology sequence. More specifically, the

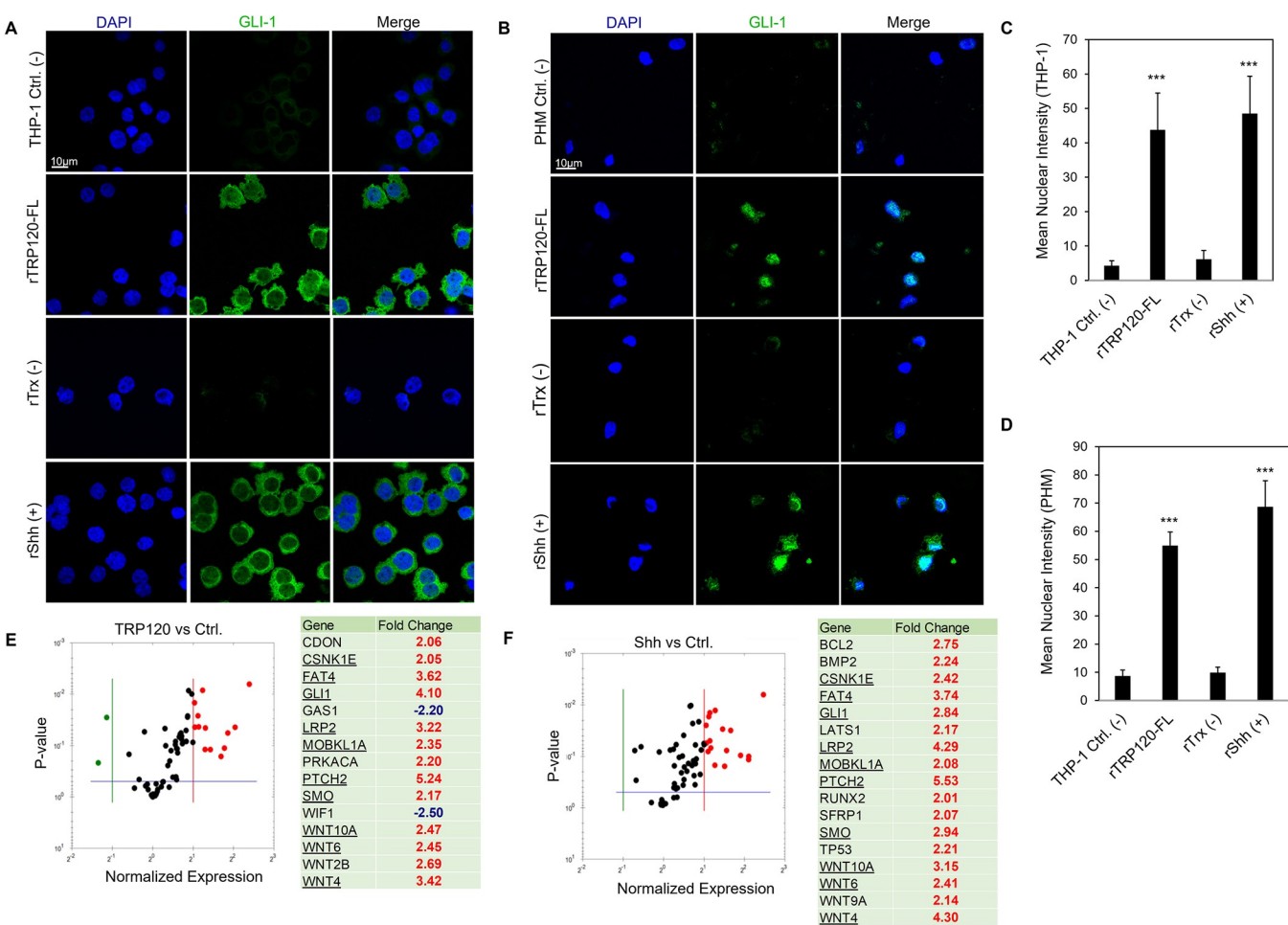

**Fig 7. *E. chaffeensis* TRP120 upregulates GLI-1 and Hh gene target expression consistent with Hh ligands.** (A) Confocal immunofluorescence microscopy of untreated (-) or rTRP120-FL-, rTrx- (-), rShh-treated (+) (1 μg/mL) THP-1 cells stained with GLI-1 antibody. Micrographs demonstrate increased levels of GLI-1 (green) in rTRP120- and rShh-treated THP-1 cells 6 h post-treatment (hpt) (scale bar = 10 μm). (B) Confocal immunofluorescence microscopy of untreated, rTRP120-FL, Trx- (-) and rShh-treated (+) (1 μg/mL) primary human monocytes harvested at 10 h. rTRP120-FL upregulation of GLI-1 (green) similar to rShh (positive control) in primary human monocytes (scale bar = 10 μm). (A-B) Experiments were performed with at least three biological and technical replicates. Randomized areas/slide (n = 10) were used to detect GLI-1 nuclear translocation. (C-D) Intensity graphs demonstrate the mean nuclear accumulation of GLI-1 in respective THP-1 cells and primary human monocytes. Analysis was performed using ImageJ and determining mean grey value from randomized areas/slide (n = 10). (E) The volcano plot is representing Hh signaling PCR array gene expression in THP-1 cells stimulated with rTRP120-FL (1 μg/mL) after normalization to control cells treated with rTrx (1 μg/mL). The respective normalized expression of rTRP120-FL regulated Hh array genes was performed with three biological and technical replicates. (F) The volcano plot is representing Hh signaling PCR array gene expression in cells stimulated with rShh (1 μg/mL) after normalized to DMSO (control) treated cells. The respective normalized expression of rShh regulated Hh array genes were performed in biological and technical replicates. (E-F) The red, black and green dots in the volcano plot represent an upregulation (≥ 2), no change and down-regulation (≤ -2), respectively for a given gene on the array. The horizontal blue lines on the volcano plots determine the level of significance (*p* ≤ 0.05).

TRP120-Hh-SLiM sequence is the sequence that was specifically defined through BLAST analysis. TRP120-Hh-SLiM-mut (18 aa) is a corresponding mutant peptide with glycine and alanine substitutions that replace the TRP120-Hh-SLiM aa's. TRP120-TR (-) (22 aa) is a TRP120 TRD sequence that does not contain the defined Hh homology sequence (**Fig 8A**). THP-1 cells were treated with TRP120-TR-Hh, TRP120-TR (-), TRP120-Hh-SLiM or TRP120-Hh-SLiM-mut for 6 h, and GLI-1 signaling was measured as described. Both TRP120-TR-Hh and TRP120-Hh-SLiM treatments elicited a significant upregulation in GLI-1, while TRP120-TR (-) and TRP120-Hh-SLiM-mut control could not upregulate GLI-1 (**Fig 8B and 8D**). Similarly, TRP120-Hh-SLiM treatment elicited a significant upregulation in GLI-1 in primary human

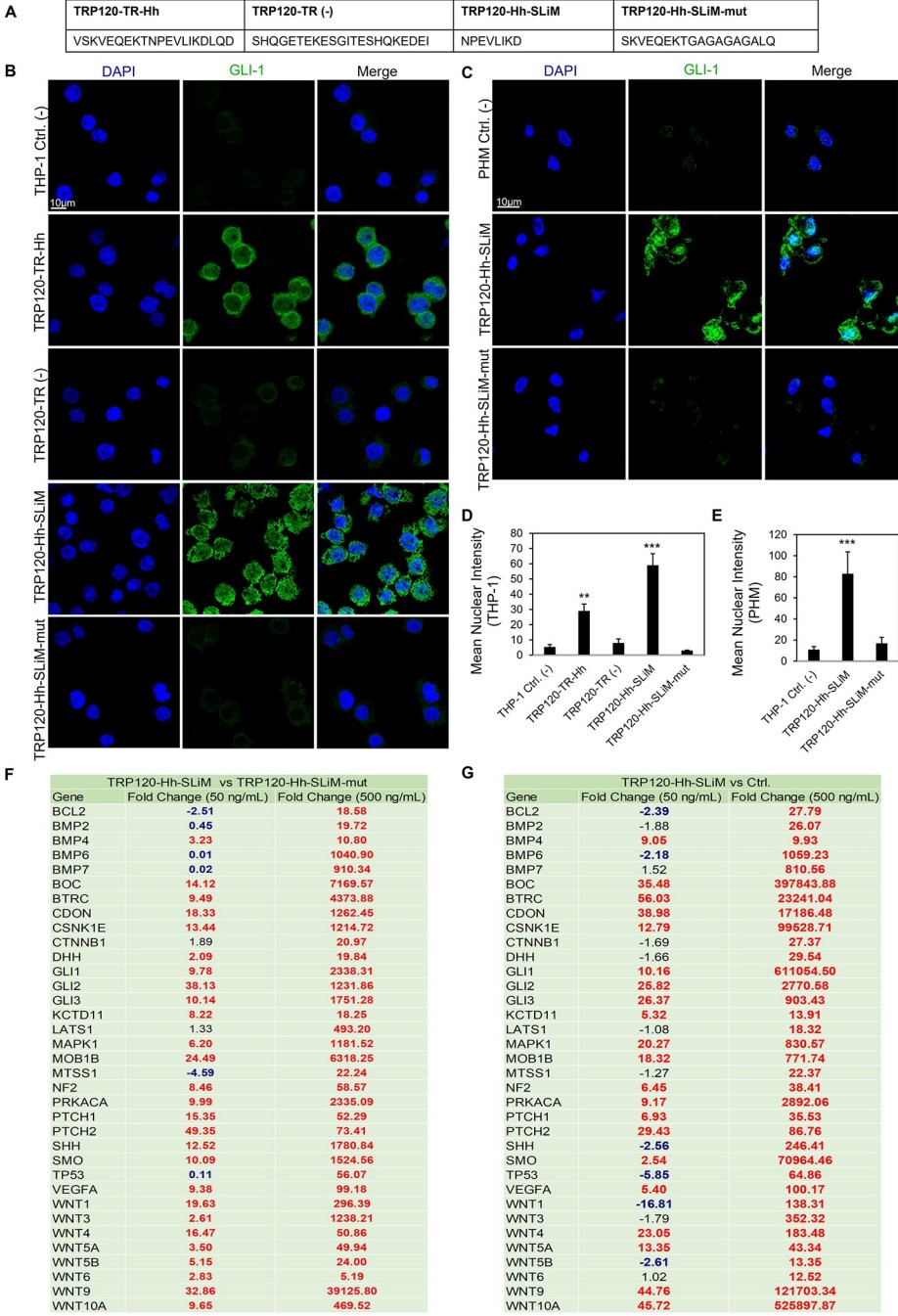

**Fig 8. TRP120-Hh-SLiM upregulates GLI-1 in THP-1 cells and primary human monocytes.** (A) TRP120-TR-Hh and TRP120-Hh-SLiM sequences contain the Hh homology sequence identified by BLAST analysis. TRP120-Hh-SLiM-mut contains glycine and alanine substitutions in the Hh SLiM region and is used as a negative control. TRP120-TR (-) is a sequence within the TRP120-TR that does not contain the defined Hh homology sequence. (B) Confocal immunofluorescence microscopy of untreated (-) or peptide treated THP-1 cells. THP-1 cells were stained with GLI-1 antibody. The micrograph shows increased levels of GLI-1 (green) in TRP120-TR-Hh and TRP120-Hh-SLiM treated, but not untreated, TRP120-TR (-) or TRP120-Hh-SLiM-mut treated THP-1 cells 6 h post-treatment (hpt) (scale bar = 10 μm). (C) Confocal immunofluorescence microscopy of untreated or SLiM/SLiM mutant peptide treated primary human monocytes harvested at 10 h. The TRP120-Hh-SLiM sequence upregulates GLI-1 (green) in primary human monocytes, but the corresponding mutant sequence does not (scale bar = 10 μm). (B-C) Experiments were performed with at least three biological and technical replicates. Randomized areas/slide (n = 10) were used to detect GLI-1 nuclear translocation. (D-E) Intensity graphs demonstrate the mean nuclear accumulation of GLI-1 in respective THP-1 cells and primary human monocytes. Analysis was performed using ImageJ and determining mean

grey value from randomized areas/slide (n = 10). (F-G) Hh signaling PCR arrays were utilized to analyze the expression of 84 Hh genes. In brief, THP-1 cells were treated with TRP120-Hh-SLiM or TRP120-Hh-SLiM-mut (50 and 500 ng/mL) or left untreated (negative control). THP-1 cells were harvested at 24 h with three biological and technical replicates. The tables represent the fold change in gene expression at each concentration. (F) The upregulation of gene expression in TRP120-Hh-SLiM-treated cells compared to untreated cells at respective concentrations. (G) Upregulation of gene expression in TRP120-Hh-SLiM treated cells compared to TRP120-Hh-SLiM-mut treated cells at respective concentrations.

monocytes at 10 hpt, but TRP120-Hh-SLiM-mut did not (**Fig 8C and 8E**). Additionally, THP-1 cells treated with 50 ng/mL or 500 ng/mL of TRP120-Hh-SLiM exhibited Hh gene regulation in a concentration dependent manner (**Fig 8F and 8G**). TRP120-Hh-SLiM significantly upregulated Hh target genes at 24 hpt compared to untreated (**Fig 8F**) and TRP120-Hh-SLiM-mut treated (**Fig 8G**) cells, including *BOC, CDON, BTRC, CSNK1E, PTCH1, PTCH2, SMO, PRKACA, LATS1, MAPK1, NF2, MTSS1, WNT10A, WNT3, WNT9a, VEGFA* and *BCL-2* as described during *E. chaffeensis* infection. These data demonstrate that the defined TRP120-Hh-SLiM activates the Hh signaling pathway and regulates Hh pathway target genes. During *E. chaffeensis* infection, we identified similar Hh activity *in vitro* using the THP-1 cell line as well as *ex vivo* with primary human monocytes. Establishing that the responses observed in the THP-1 cells are similarly observed in primary human monocytes cultured *ex vivo*, which is important because of the limited lifespan of primary cells and the advantages of using THP-1 cell line for functional laboratory studies that may serve as a foundation for understanding mechanisms and potential therapeutics that could be used for treatment in patients.

## A TRP120-Hh-SLiM targeted antibody blocks Hh signaling

To elucidate the role of the TRP120-Hh-SLiM during *E. chaffeensis* infection, we investigated whether blocking *E. chaffeensis* infection or the TRP120-Hh-SLiM with a TRP120-Hh-SLiM targeted antibody would inhibit Hh signaling. We used a neutralization assay to determine antibody effects on Hh signaling during *E. chaffeensis* infection or TRP120-Hh-SLiM treatment. *E. chaffeensis* or TRP120-Hh-SLiM were incubated with 1.5 μg/mL of either α-TRP120-I1 antibody (targets TRP120 sequence SKVEQEETNPEVLIKDLQDVAS) or α-TRP32 antibody (control) for 1 h or overnight, respectively, and then THP-1 cells were subsequently treated with each mixture for 10 h. *E. chaffeensis* infected and TRP120-Hh-SLiM treated cells in the presence of α-TRP120-I1 demonstrated significant reduction in GLI-1 expression relative to *E. chaffeensis*-infected and TRP120-Hh-SLiM treated cells in the presence of α-TRP32 antibody (**Fig 9A–9C**). These data confirm that the TRP120-Hh-SLiM upregulates GLI-1 and the interaction can be blocked by antibody.

## *E. chaffeensis* TRP120-Hh-SLiM upregulates BCL-2 expression

During gene expression analysis of the Hh signaling pathway, we observed a significant increase in *BCL-2* gene transcription. *BCL-2* is one of the major transcriptional targets of the Hh signaling pathway [25]. BCL-2 is involved in maintaining mitochondrial membrane integrity and preventing activation of caspases by inhibiting cytochrome-c release from mitochondria, thus inhibiting the intrinsic apoptotic pathway [47]. Hence, we hypothesized that *E. chaffeensis* activates Hh to upregulate BCL-2 to engage an anti-apoptotic cellular program. *E. chaffeensis*-infected (**Fig 10A**)**,** TRP120-Hh-SLiM- or TRP120-Hh-SLiM-mut-treated (**Fig 10B**) THP-1 cells were collected for immunoblot to determine BCL-2 levels. Significant upregulation of BCL-2 was detected in *E. chaffeensis*-infected and TRP120-Hh-SLiM-treated cells compared to negative controls. Further, we examined the effect of BCL-2 inhibition on *E.*

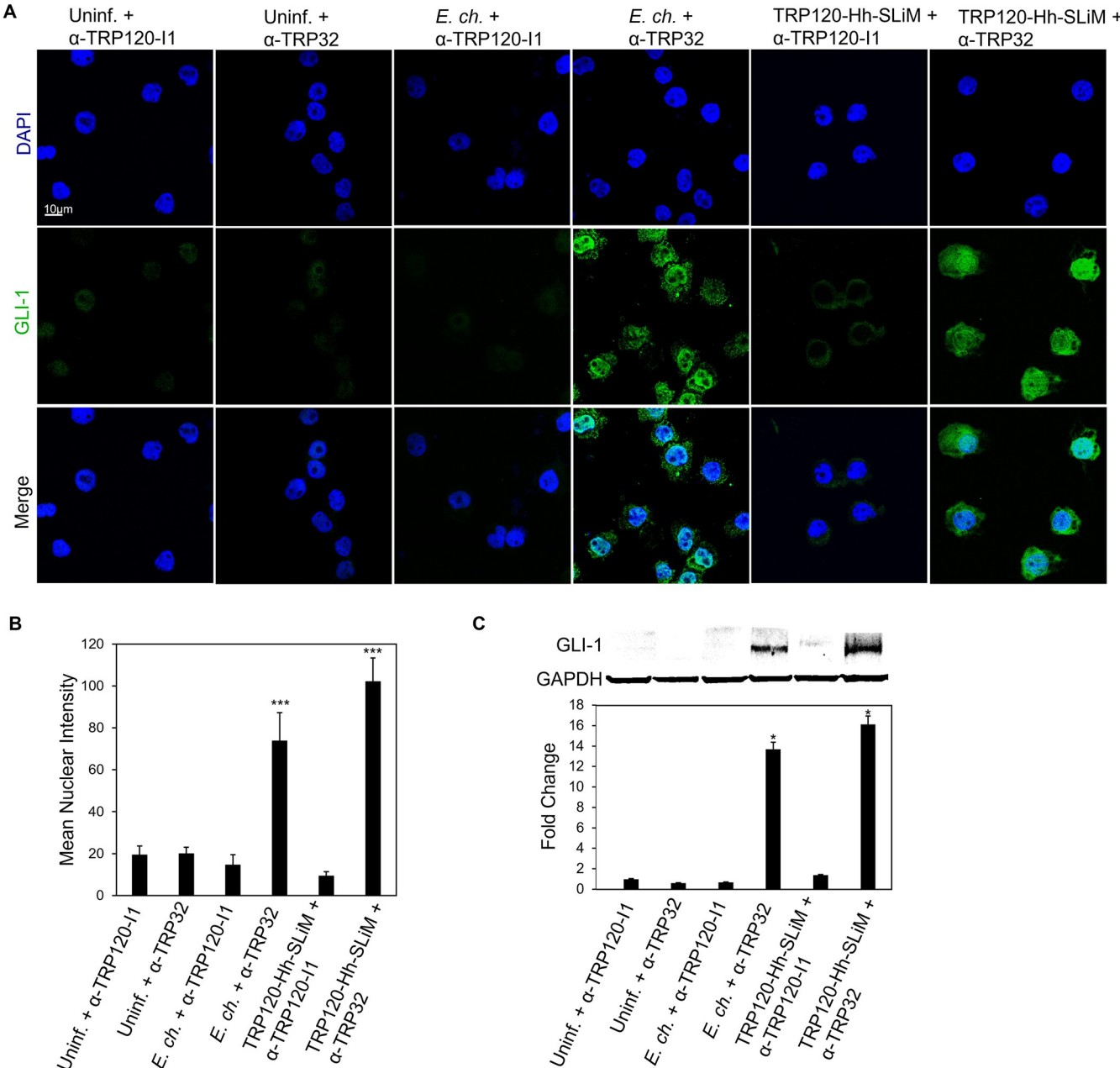

**Fig 9. Anti-TRP120-Hh-SLiM antibody blocks Hh signaling and GLI-1 nuclear translocation.** (A) *E. chaffeensis* (MOI 100) and SLiMs were incubated with α-TRP120-I1 (targets TRP120 sequence SKVEQEETNPEVLIKDLQDVAS) or α-TRP32 (neg ctrl) (1.5 μg/mL) for 1 h or overnight, respectively, before incubation with THP-1 cells. THP-1 cells were harvested at 10 hpt, immunostained with GLI-1 (green), and visualized by confocal fluorescence microscopy. Scale bar = 10 μm. 10 randomized areas/slide were used to detect GLI-1 nuclear translocation. (B) Intensity graph demonstrates the mean nuclear accumulation of GLI-1 in respective THP-1 cells. Analysis was performed using ImageJ and determining mean grey value from randomized areas/slide (n = 10). (C) Western blot analysis of treatment groups with GAPDH as a loading control. Data are represented as means ± SD (*$p < 0.05$). (A-B); α-TRP120-I1 inhibits GLI-1 upregulation in cells with *E. chaffeensis* or TRP120-Hh-SLiM compared to α-TRP32. Untreated cells were incubated with α-TRP120-I1 or α-TRP32 as negative controls. Experiments were performed with at least three biological and technical replicates and significance was determined through *t*-test analysis. Randomized areas/slide (n = 10) were used to detect GLI-1 nuclear translocation.

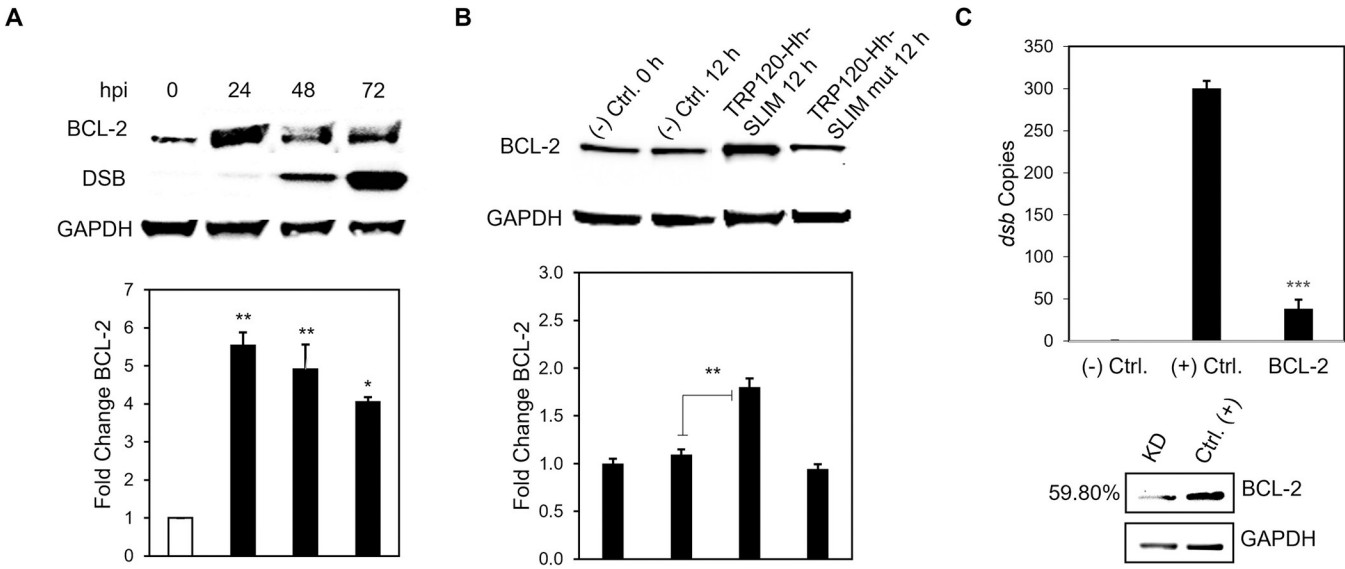

**Fig 10. *E. chaffeensis* deploys the TRP120-Hh-SLiM to induce BCL-2 expression for survival.** (A) Western blot analysis of BCL-2 levels of *E. chaffeensis* infected (MOI 100) THP-1 cells collected at 0, 24, 48 and 72 hpi with GAPDH as a loading control and Dsb as an infection control. *E. chaffeensis* induces BCL-2 protein expression. (B) BCL-2 levels of TRP120-Hh-SLiM and TRP120-Hh-SLiM-mut treated and untreated THP-1 cells collected at 12 hpt with GAPDH as a loading control. The TRP120-Hh-SLiM peptide induces BCL-2 protein expression, but the mutant does not. (B, C) Bar graphs depict Western blot densitometry values normalized to GAPDH. (C) siRNA knockdown (KD) of BCL-2 significantly reduces *E. chaffeensis* infection in THP-1 cells as determined by qPCR amplification of the *dsb* gene. Scrambled siRNA (scrRNA) was transfected for positive (infected) and negative control (mock infected). Western blot depicts knockdown efficacy of siRNA in knockdown cells compared to the positive control. Number left of siRNA lane indicates percent knockdown of BCL-2 relative to positive control, normalized to GAPDH expression. (A-C) Experiments were performed with at least three biological and technical replicates and significance was determined through *t*-test analysis. Data are represented as means ± SD (*$p < 0.05$; **$p < 0.01$; ***$p < 0.001$).

*chaffeensis* infection using siRNA to individually target and silence *BCL-2* in THP-1 cells. At 24 hpi of *BCL-2* siRNA-transfected cells, *E. chaffeensis* infection (depicted by *dsb* copy number) was significantly reduced (**Fig 10C**). Collectively, these data suggest that *E. chaffeensis* utilizes its TRP120-Hh-SLiM to activate the Hh signaling pathway and upregulate BCL-2 for intracellular survival.

## *E. chaffeensis* mediated activation of Hh signaling inhibits host cell apoptosis

It is well documented that Hh signaling promotes cell proliferation and prevents cell apoptosis through BCL-2 activation [24,25,45]. BCL-2 is involved in the inhibition of mitochondria-mediated pro-death pathway [48]. Based on our results demonstrating the importance of BCL-2 in *E. chaffeensis* infection, we hypothesized that *E. chaffeensis* activates Hh signaling to inhibit mitochondria-mediated host cell apoptosis via activation of Hh signaling. To examine this hypothesis, infected and uninfected THP-1 cells were treated with Etoposide, an inhibitor of topoisomerase II and inducer of cellular apoptosis, and stained with the JC-1 dye (**Fig 11A**). *E. ch.*-infected Etoposide-treated THP-1 cells exhibited a significant increase in cells with JC-1 aggregates, suggesting active inhibition of host cell apoptosis during *E. chaffeensis* infection compared to uninfected Etoposide-treated cells. Additionally, there were significantly fewer apoptotic cells in DMSO and Etoposide + *E. chaffeensis* groups, but significantly more apoptotic cells in Etoposide groups, suggesting that *E. chaffeensis* inhibits apoptosis in the presence of Etoposide (**Fig 11B**). We further confirmed the loss of mitochondrial membrane potential in the presence of SMO-specific inhibitor Vismodegib in *E. chaffeensis*-infected cells using JC-1 dye. The micrograph shows the presence of mitochondria with positive membrane potential in

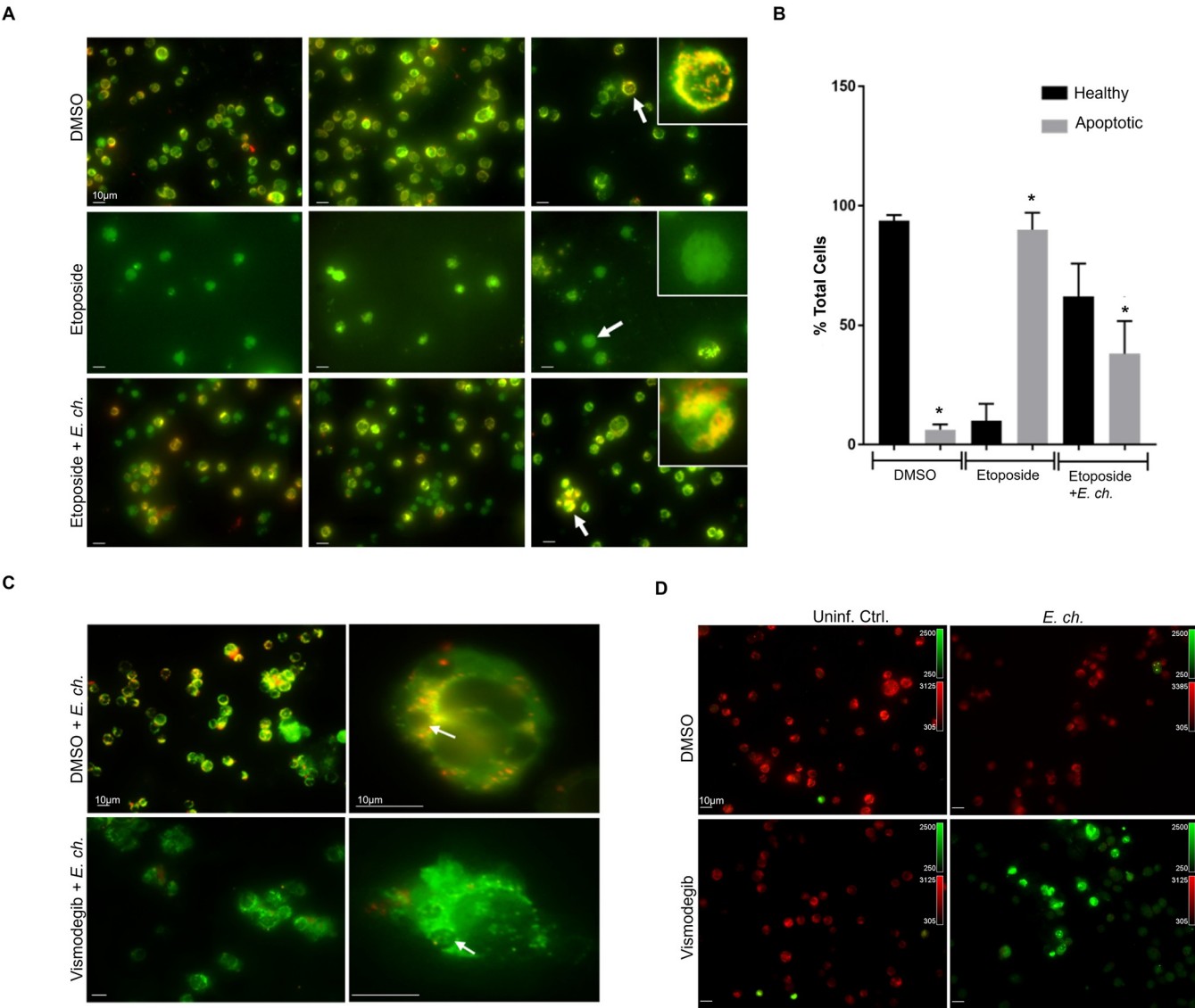

**Fig 11. *E. chaffeensis* mediated activation of Hh signaling inhibits host cell apoptosis.** (A) Immunofluorescence analysis of uninfected and *E. chaffeensis*-infected THP-1 cells stained with JC-1 dye after treatment with 100 µM of Etoposide or DMSO control. The micrographs demonstrate the formation of JC aggregates (orange; 590±17.5 nm) in mitochondria with positive membrane potential in DMSO treated cells (top panel). Due to depolarization of the mitochondrial membrane in Etoposide-treated cells, JC-1 remains as a monomer and yields green fluorescence with the emission of 530±15 nm (middle). *E. chaffeensis*-infected (MOI 100) Etoposide-treated THP-1 cells showed increased cells with JC-1 aggregates, indicating active inhibition of host cell apoptosis during *E. chaffeensis* infection (bottom). (B) Bar graph depicts percentage of normal or apoptotic cells in DMSO, Etoposide, or Etoposide + *E. chaffeensis* groups. Percent of total cells were counted from randomized areas/slide (n = 10). There were significantly fewer apoptotic cells in DMSO and Etoposide + *E. chaffeensis* groups, but significantly more apoptotic cells in Etoposide treated groups. Experiments were performed in biological and technical replicates and significance was determined through *t*-test analysis. Data are represented as means ± SD (*$p<0.05$). (C) Immunofluorescence analysis of mitochondrial membrane potential using JC-1 dye in *E. chaffeensis*-infected (MOI 100) THP-1 cells treated with Hh inhibitor Vismodegib or DMSO. The micrograph demonstrated the presence of mitochondria with positive membrane potential in DMSO treated infected THP-1 cells compared to Vismodegib-treated infected (MOI 100) THP-1 cells. Arrow points to the ehrlichial inclusion. (D) Immunofluorescence analysis of uninfected and *E. chaffeensis*-infected (MOI 100) THP-1 cells stained with Nucview488 and the Mitoview633 Dye after treatment with Vismodegib or DMSO control. The micrographs show Nucview488 dye is cleaved by caspase 3 and produces green fluorescence in Vsmodegib-treated *E. chaffeensis*-infected cells due to the activation of apoptosis. In comparison, due to positive mitochondrial membrane potential, Mitoview633 accumulated in the inner mitochondrial membrane (red) in Vismodegib-uninfected and DMSO treated (Ctrl) uninfected and infected cells. These results demonstrate that Hh signaling plays a crucial role during *E. chaffeensis* infection by inhibiting intrinsic death signaling. (A, C, D) Experiments were performed with at least three biological and technical replicates. Randomized areas/slide (n = 10) were selected to visualize the phenomenon. Scale bar = 10 µm.

DMSO-treated cells infected with *E. chaffeensis* compared to Vismodegib-treated cells infected with *E. chaffeensis* (**Fig 11C**). Additionally, we treated cells with Vismodegib or DMSO and examined cellular apoptotic state using the Nucview488 and the Mitoview 633 apoptosis assay. During *E. chaffeensis* infection and in the presence of Vismodegib, the Nucview488 dye (a substrate of active caspase 3) translocated to the nucleus, producing green fluorescence in infected cells compared to uninfected cells in the presence of Vismodegib. We did not observe any measurable difference in DMSO-treated uninfected and infected cells (**Fig 11D**). These results demonstrate that Hh signaling plays a crucial role during *E. chaffeensis* infection in monocytes by inhibiting an intrinsic death signal.

### *E. chaffeensis* induces an apoptotic profile in the presence of Hh inhibitor

Our results demonstrate the importance of anti-apoptotic protein BCL-2 during *E. chaffeensis* infection. Additionally, we reveal the vital role that Hh signaling plays during *E. chaffeensis* infection to inhibit apoptosis. Based on our data demonstrating increased Nucview 488 dye in the nucleus of *E. chaffeensis* infected cells in the presence of Vismodegib, we hypothesized that *E. chaffeensis* activates Hh signaling, thus upregulating BCL-2 to inhibit caspases 9 and 3. To confirm our hypothesis, *E. chaffeensis*-infected and uninfected THP-1 cells were treated with Vismodegib or DMSO (**Fig 12A**). *E. chaffeensis*-infected Vismodegib-treated THP1 cells demonstrated a significant increase in cytoplasmic condensation (precursor to apoptosis) at 24 hpi compared to uninfected Vismodegib-treated cells and *E. chaffeensis*-infected and uninfected DMSO-treated cells, supporting the conclusion that *E. chaffeensis* activates Hh signaling to prevent apoptosis. Additionally, ehrlichial survival was significantly reduced in the presence of Vismodegib compared to DMSO (**Fig 12B**). Further, cell viability significantly decreased in *E. chaffeensis*-infected cells treated with Vismodegib (**Fig 11B**). To define a direct mechanism by which *E. chaffeensis* activates Hh signaling to prevent apoptosis, we evaluated levels of BCL-2, caspase 9 and caspase 3 (**Fig 12D–12F**). *E. chaffeensis*-infected Vismodegib-treated THP-1 cells showed a significant decrease in BCL-2 at 24 hpi compared to infected DMSO-treated cells (**Fig 12D**). In addition, *E. chaffeensis*-infected Vismodegib-treated THP1 cells show a significant decrease in pro caspases 3 and 9 and a significant increase in cleaved caspases 3 and 9 at 24 hpi compared to infected DMSO-treated cells (**Fig 12E and 12F**). Collectively, these results define a direct mechanism by which *E. chaffeensis* targets Hh signaling to induce BCL-2 expression, thus preventing intrinsic apoptosis.

## Discussion

The Hh pathway was first identified in *Drosophila* in the 1970s and for the last couple of decades it has mostly been studied in the field of developmental biology [49]. More recent investigations have shown Hh role in cell proliferation, differentiation, and inhibition of apoptosis and unregulated activation of Hh signaling results in different hematological malignancies and other cancerous conditions [22,24,50,51]. The Hh pathway is targeted by multiple pathogens [19,20]; however, the specific mechanism by which pathogens target Hh signaling through SLiMs has not been reported. In this study, we identified a Hh SLiM within the *E. chaffeensis* TRP120 effector that activates the Hh signaling pathway and inhibits intrinsic host-cell apoptosis to enable infection of the monocyte. This is the first report of a eukaryotic Hh SLiM mimetic in bacterial proteomes, which represents a novel virulence strategy by obligate intracellular bacteria and extends knowledge regarding eukaryotic cellular signaling motifs that are relevant in pathogen-host interplay. This investigation and other recent reports from our laboratory have provided compelling detail of the molecular mechanisms that *E. chaffeesis* uses to reprogram the host cell using SLiM mimicry. We have identified an array of eukaryotic

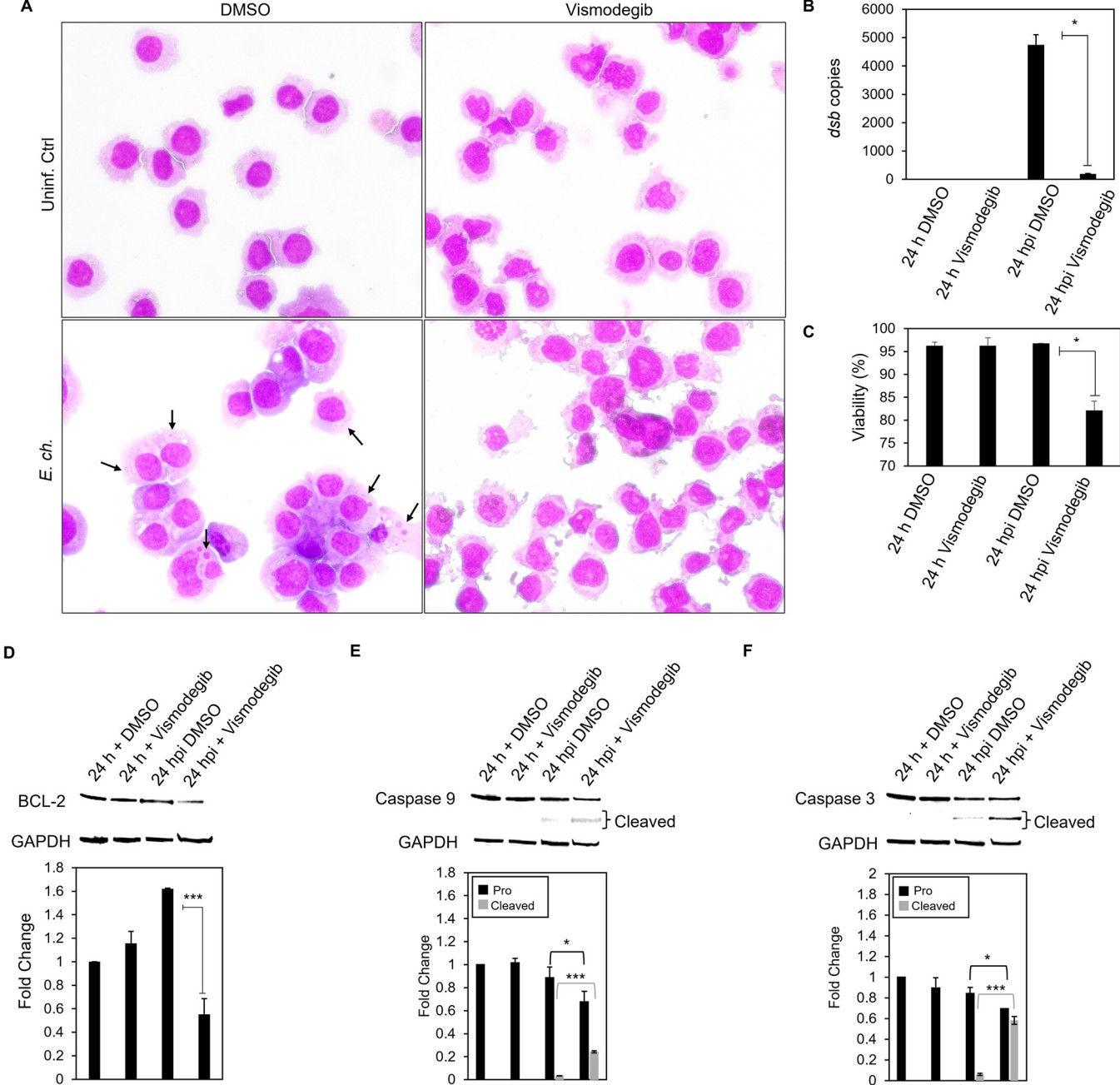

**Fig 12. Apoptotic profile is induced in the presence of Hh inhibitor during *E. chaffeensis* infection.** (A) Brightfield micrographs showing effects of DMSO or Hh inhibitor Vismodegib on uninfected and *E. chaffeensis* infected THP-1 (MOI 50) cells prepared using Diff-Quick staining. THP-1 cells were treated with Vismodegib or DMSO (200 nM) and infected with *E. chaffeensis* or uninfected for 24 h. Infected THP-1 cells treated with Vismodegib undergo cytoplasmic condensation (precursor to apoptosis), but other treatment groups do not (arrows point to morulae). (B) Bar graph showing fold-change in *E. chaffeensis* infection for each treatment group. Ehrlichial loads were determined using qPCR measurement of *dsb* copy and normalized with host cell *GAPDH*. *E. chaffeensis* infection significantly declines in the presence of Vismodegib. (C) Bar graphs showing cell viability for each treatment group. Cell viability was determined using the Cellometer Mini bright field imaging and pattern-recognition assay. Cell viability significantly declines in the presence of Vismodegib during *E. chaffeensis* infection. (D) Western blot analysis of BCL-2 levels for each group with GAPDH as a loading control. BCL-2 protein expression significantly declines during *E. chaffeensis* infection in the presence of Vismodegib. (E) Western blot analysis of pro and cleaved caspase 9 levels for each group with GAPDH as a loading control. Pro caspase 9 protein expression significantly declines while cleaved caspase 9 protein expression significantly increases during *E. chaffeensis* infection in the presence of Vismodegib. (F) Western blot analysis of pro and cleaved caspase 3 levels for each group with GAPDH as a loading control. pro caspase 3 protein expression significantly declines while cleaved caspase 3 protein expression significantly increases during *E. chaffeensis* infection in the presence of Vismodegib. (D-F) Bar graphs depict Western blot densitometry values normalized to GAPDH. (B-F) (A-F) Experiments were performed with at least three biological and technical replicates and significance was determined through *t*-test analysis. Data are represented as means ± SD (*$p<0.05$; **$p<0.01$; ***$p<0.001$).

ligand SLiMs positioned in a single surface expressed effector protein that interface with the host cell and activate Notch, Wnt and now Hh signaling to counter innate defense mechanism and promote infection [3,5]. Collectively, these studies provide the molecular basis of eukaryotic pathway activation by an intracellular pathogen and provides a model that is valuable for understanding how pathogens interface with eukaryotic cells and rewire host cell pathways for infection.

The Hh pathway has been implicated in various human diseases including several types of cancers [22,24,42]. Notably, several recent studies have reported increased levels of Hh signaling in response to various pathogens [20,52–54]. An elevated level of Hh signaling was reported in cells infected with Hepatitis B and C virus (HBV and HCV, respectively), and hepatocytes from patients with chronic HBV and HCV infection displayed an increased production of Hh ligands and an accumulation of Hh-responsive cells with higher levels of pathway activity [52]. Additionally, studies demonstrate that the *in vitro* treatment of hepatocytes with whole HBV replicon increases expression of Hh target genes in a GLI-dependent manner and viral protein HBV X stabilizes GLI-1 and promotes its nuclear accumulation [54]. Although a precise mechanism by which Hh signaling promotes HBV and HCV infections remains unclear, Hh activation in hepatocytes appears to promote HCV infection, indicating the presence of a positive feedback loop between pathway activation and virus production [55]. In addition, *Mycobacterium* species appear to mediate Hh signaling. *M. bovis* upregulates Shh-PI3K-mTOR-NF-κB signaling in human dendritic cells to activate BCG-induced Treg expansion. Interestingly, *M. bovis* relies heavily on Hh signaling, while Notch signaling hindered the ability of the infected dendritic cells to expand Tregs and Wnt signaling demonstrated no affect [53].

*Ehrlichia chaffeensis* TRP120 is a moonlighting protein involved in modulating various cellular processes [6] and has evolved Wnt and Notch SLiMs to activate the conserved cellular signaling pathways [3,5]. In a recent report demonstrating Notch activation during *E. chaffeensis* infection, we determined *GLI-1* gene expression was activated by a Notch ligand SLiM in the TRP120 TR [5,36]. The intricate crosstalk between Wnt, Notch, and Hh signaling is well known; thus, we investigated the possibility of additional ligand SliMs and Hh pathway activation during *E. chaffeensis* infection [37]. Utilizing similar *in silico* analysis, a potential TRP120-Hh-SLiM was predicted within the TRP120 TRD with significant homology to a region of Hh ligands at the Hh ligand-PTCH receptor binding site [39]. Using iRNA, we demonstrated that *E. chaffeensis* relies on PTCH2, but not PTCH1 for survival. Although PTCH1 and PTCH2 share overlapping functions [56], PTCH1 contains an ubiquitin ligase binding site within its C-terminal tail, making it less stable than PTCH2, which may make PTCH1 less favorable to TRP120 [57]. We determined that TRP120 TRD interacts directly with PTCH2 loop 1 region and has a binding affinity in the nM range. Moreover, multiple studies have investigated the affinity of Hh ligands to PTCH receptors and reported ligand affinity for PTCH1 (Shh, 1.0 nM; Dhh, 2.6 nM; Ihh, 1.0 nM) and PTCH2 (Shh, 1.8 nM; Dhh, 0.6 nM; Ihh, 0.4 nM) [58]. Interestingly, Shh binds PTCH2 with higher affinity, while Dhh and Ihh bind PTCH1 with higher affinity. TRP120 may bind PTCH2 similarly to Shh in our model, since Shh is highly expressed in THP-1 cells compared to Dhh and Ihh. Further, the affinity of endogenous Hh ligands for PTCH2 is lower than what was exhibited by TRP120 (4.40 ± 1.5 nM). A recent study demonstrates that ligands with higher binding affinity disable lower affinity ligands from binding their receptor [59]. Thus, TRP120 may have a higher binding affinity to PTCH2 to provide a competitive advantage over endogenous Hh ligands.

We investigated whether TRP120 is directly responsible for activating Hh signaling during infection. Indeed, we confirmed that TRP120 induces nuclear translocation of GLI-1 and transcriptional induction of Hh pathway genes including crucial components of the Hh-signaling

pathway such as PTCH2, SMO, and GLI-1. Although there was differential expression of Hh pathway genes in TRP120 treated and Shh ligand treated THP-1 cells, we discovered that more than 50% of genes were common during TRP120 and Shh treatment, which supports TRP120 as a Hh ligand mimic. Differences between TRP120 and Shh are expected, since there are differing biological functions between Shh, Dhh and Ihh ligands despite a highly similar amino acid sequence [60]. Additionally, TRP120 also contains Wnt and Notch SLiMs which may influence gene expression due to the intricate crosstalk between the pathways. For example, Notch signaling directly regulates effector and target molecules of the Hh signaling pathway [61]. Additionally, Wnt signaling increases GLI-1 transcriptional activity [62], which suggests that TRP120 can regulate Hh signaling in various ways. In addition, we concluded that the predicted TRP120-Hh-SLiM peptide is sufficient in activating Hh signaling. Here, we established a model of eukaryotic protein mimicry where the TRP120-Hh-SLiM is functional and activates the Hh signaling pathway and Hh gene targets in a concentration dependent manner. In our experiments, the TRP120-Hh-SLiM demonstrated stronger upregulation of Hh gene targets than full length TRP120. This is likely related to the molar concentration of SLiM sequences present in each treatment. The actual amount of SLiM added using rTRP120 is substantially less than the concentration of peptide SLiM used. Nevertheless, there were similar and consistent Hh gene activation profiles observed with *E. chaffeensis*, TRP120 and TRP120-Hh-SLiM.

The TRP120-Hh-SLiM has sequence homology with the N-terminal region of Shh, which is responsible for binding PTCH receptors [63]. However, the amino acid residues important for Shh-PTCH2 interactions have not been well defined. Our findings demonstrate that mutations in the homology sequence between TRP120 and Hh ligands results in deactivation of Hh signaling, which indicates that these amino acid residues may be critical in Hh ligand-PTCH2 binding and subsequent GLI-1 activation. To further support our results, we used an antibody that would recognize and block the TRP120-Hh-SLiM. Indeed, we determined that antibody binding and blocking of the Hh SLiM inhibits both *E. chaffeensis* and TRP120-Hh-SLiM activation of Hh signaling. These experiments also demonstrate that the TRP120-Hh-SLiM is the only Hh mimietic utilized by *E. chaffeensis*, since the antibody blocked GLI-1 upregulation in SLiM treated and *E. chaffeensis* infected groups.

In the past year, our laboratory has demonstrated that TRP120 contains multiple SLiMs that activate Hh, Notch and Wnt signaling, which likely work together to promote infection due to the well-known but complex crosstalk between Hh, Notch and Wnt pathways [61]. Each TRP120 SLiM is found within the intrinsically disordered TRD within close proximity, suggesting that the TRD has a primary and potentially unique role in SLiM mimicry [3,64]. TRDs are often identified in bacterial proteins with critical functions for pathogenicity [65]. For example, *Xanthomonas* secreted TAL effectors contain repeats, which facilitate DNA binding and gene regulation [66,67]. Similarly, we have identified DNA binding capability in the TR domain of TRP120 [12]. SLiMs drive evolution and rapidly evolve *ex nihilo* to add new functionality to proteins. Pathogens are known to convergently evolve SLiMs within disordered regions due to the limited number of mutations necessary for the generation of a new SLiM [68]. SLiMs contain high evolutionary plasticity due to their disordered nature, short length and limited number of specificity-determining residues [69]. Thus, it appears that *E. chaffeensis* has evolved TRP120 SLiMs through convergent evolution to increase complexity necessary for engaging multiple cellular signaling pathways. All presently defined SLiMs acquired by TRP120 activate conserved signaling pathways known to prevent apoptosis, which may be a strategy utilized by *E. chaffeensis* to assure host-cell survival for the *Ehrlichia* life cycle.

During microbial infection, cellular apoptosis plays an important role as a host defense mechanism, as it minimizes infection and contributes to protective immunity through

processing apoptotic bodies containing infected microbes to facilitate antigen presentation (31). Intracellular bacteria usually require several days of replication in a host cell before being released to infect neighboring cells. Thus, pathogens like *Mycobacterium*, *Chlamydia*, *Rickettsia*, *Anaplasma*, and others have evolved multiple mechanisms to inhibit host cell apoptosis [70–74]. Intracellular pathogens regulate host cell apoptosis to modulate the host immune defenses in a variety of ways, including regulation of the mitochondria-mediated intrinsic apoptosis pathway [29,30,75]. For instance, *E. chaffeensis* utilizes the Type IV secretion system (T4SS) effector Etf-1 to enter the mitochondria and inhibit mitochondria-mediated intrinsic apoptosis in host cells [76]. T4SS effectors are well known for their virulence role in preventing apoptosis during infection [77]. However, there are many mechanisms pathogens exploit to inhibit apoptosis. Notably, in this study we demonstrate a novel mechanism associated with a T1SS effector capable of inhibiting apoptosis. We reveal that TRP120 significantly increases BCL-2 expression via its Hh SLiM. Additionally, siRNAs against *BCL-2* significantly reduced ehrlichial load, suggesting that Hh regulated BCL-2 anti-apoptotic mechanisms are an important component of an anti-apoptotic strategy by *Ehrlichia*. Notably, *in vitro* studies demonstrate the upregulation of BCL-2 in host macrophages during *M. tuberculosis* infection to prevent apoptosis for intracellular survival [78]. Additionally, *M. tuberculosis* secretes PtpA to dephosphorylate host protein GSK3 and suppresses caspase 3 during early infection to prevent host-cell apoptosis [79]. Interestingly, GSK3 is well known for its role during Hh signaling by regulating GLI [80].

Hh signaling plays a critical role in promoting differentiation, proliferation and maturation, and preventing apoptosis of different immune cells, including monocytes and macrophages [81,82]. A small molecule Hh inhibitor of the cell surface receptor SMO (Vismodegib) confirmed that the Hh pathway inhibits apoptosis and is required for ehrlichial survival. We confirmed that loss of Hh signaling during ehrlichial infection induces the intrinsic apoptotic pathway. Specifically, we conclude that *E. chaffeensis* upregulates BCL-2 to prevent intrinsic apoptosis by maintaining the integrity of the mitochondrial membrane, preventing the release of cytochrome c, and activation of caspase 9 and caspase 3. These results reveal a novel mechanism by which *E. chaffeensis* modulates the Hh pathway for infection by extending the host cell lifespan, which is consistent with the role of this pathway in cell biology.

Diving deeper into understanding the molecular mechanisms of *E. chaffeensis* pathogenesis in modulating complex cellular processes will help in developing next-generation therapeutics, specifically aimed at mechanistically defined host targets. The current study reveals a novel mechanism where *E. chaffeensis* utilizes SLiM ligand mimicry to activate Hh signaling in the host, thus modulating the intrinsic apoptotic signal as depicted in **Fig 13**. Hence, this study reveals the importance of the Hh signaling pathway in ehrlichial intracellular growth and developmental cycle and provides a new target for the development of a novel therapeutic approach against ehrlichial infection that may be applicable to other intracellular pathogens, in which exploitation of such conserved cellular pathways is necessary for infection.

## Materials and methods

### Cell culture and *E. chaffeensis* cultivation

Human monocytic leukemia cells (THP-1; ATCC TIB-202) or primary human monocytes (PHMs) were propagated in RPMI 1640 with L-glutamine and 25 mM HEPES buffer (Invitrogen, Carlsbad, CA), supplemented with 1 mM sodium pyruvate (Sigma-Aldrich, St. Louis, MO), 2.5 g/liter D-(+)-glucose (Sigma-Aldrich), and 10% fetal bovine serum at 37°C in a 5% $CO_2$ atmosphere. Human primary monocytes were isolated using MACS negative selection (Miltenyi Biotec, Cambridge, MA) from peripheral blood mononuclear cells obtained from

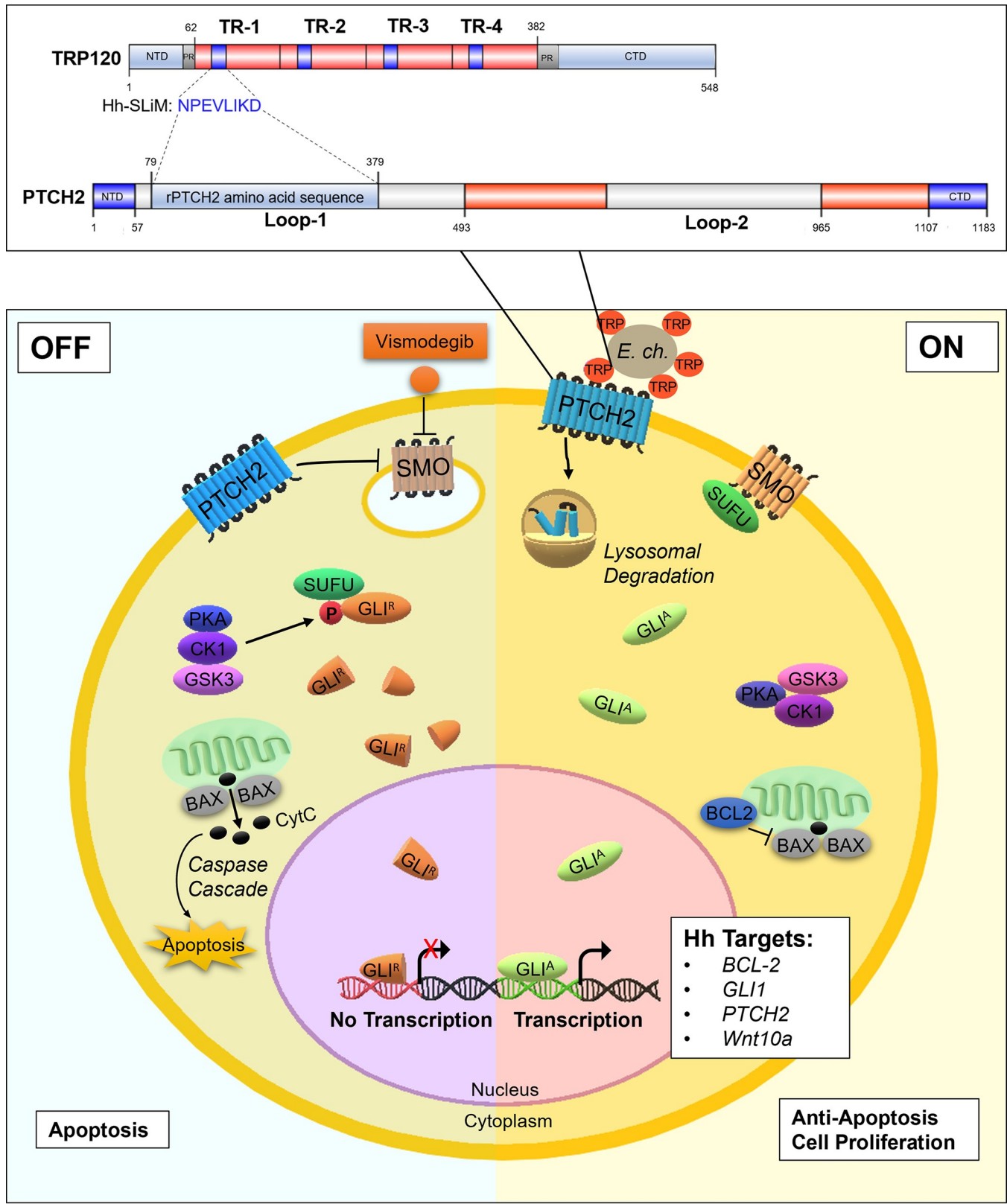

**Fig 13. Model of *E. chaffeensis* TRP120 SLiM mimetic activation of Hh signaling, downstream GLI-1 upregulation and apoptosis inhibition.** When Hh signaling is off, GLI-1 is negatively regulated by SUFU, which leads to GLI-1 phosphorylation and truncation. Truncated GLI-1 (GLI^R) translocates to the

nucleus, but its repressive form is unable to upregulate downstream gene targets. Thus, BCL-2 is not expressed, which leads to Bax release of CytC and subsequent activation of caspase cleavage and apoptosis. Therefore, *E. chaffeensis* TRP120 SLiM interacts with the PTCH2 receptor loop 1 region to trigger PTCH2 lysosomal degradation, thereby activating the SMO receptor to prevent SUFU from inhibiting GLI-1. Therefore, transcription factor GLI-1 (GLI^A) translocates freely to the nucleus to bind DNA and upregulate Hh gene targets. Further, Hh signaling upregulates BCL-2 to prevent apoptosis by maintaining the integrity of the mitochondrial membrane and thus prevents Bax release of cytochrome c (CytC) and involved activation of the intrinsic caspase cascade (caspase 9, caspase 3).

healthy human donors (de-identified) (Gulf Coast Regional Blood Center, Houston, TX)(5). *Ehrlichia chaffeensis* (Arkansas strain) was cultivated in THP-1 cells as previously described [83]. Cells were harvested with 30% confluency for confocal microscopy and 100% confluency for all other experiments.

## Protein sequence analysis

The TRP120 protein sequence (NCBI gene accession number AAO12927.1) and Dhh/Ihh/Shh *Homo sapiens* protein sequences (NCBI gene accession numbers NP_066382/ NP_002172/ NP_000184.1) were analyzed by the NCBI Protein Basic Local Alignment Search Tool (Protein BLAST) for sequence alignment.

## Informational spectrum method analysis

Informational spectrum method (ISM) *in-silico* analysis was performed by the Biomed Protection (biomedprotection.com) using a wEB platform and described in detail previously [3].

## Recombinant proteins and peptides

*E. chaffeensis* recombinant full length TRP120 (rTRP120-FL), TRP120 TRD (rTRP120-TR) or thioredoxin (rTrx; ctrl) were expressed in *E. coli* and purified as described previously [36]. rTRP120 is a Trx-fusion protein; therefore, rTrx was used as a negative control. rShh (R&D Systems, Minneapolis, MN) and rPTCH2 (MyBioSource, San Diego, CA) were obtained from a commercial source. rPTCH2 (amino acids 79 to 379; putative loop 1 ligand binding region) was expressed in yeast and contains a N-terminal 6xHis-tag. Shh was selected as a positive control since it is normally expressed in THP-1 cells, while Dhh and Ihh are not (The Human Protein Atlas; www.proteinatlas.org). Peptides were commercially synthesized (GenScript, Piscataway, NJ) for TRP120-TR-Hh (VSKVEQEKTNPEVLIKDLQD; contains the homologous Hh sequence), TRP120-TR (-) (SHQGETEKESGITESHQKEDEI; neg ctrl), TRP120-Hh-SLiM (NPEVLIKD) and TRP120-Hh-SLiM-mut (SKVEQEKTGAGAGAGALQ; Gly/Ala substitutions in the Hh SLiM motif).

## Antibodies and inhibitors

Antibodies used in this study include α-disulfide bond formation protein (Dsb) [84], α-TRP120-I1 (targets TRP120 sequence SKVEQEETNPEVLIKDLQDVAS) [85], α-TRP32 [86], α-GLI-1/2/3 (Santa Cruz Biotechnology, Dallas, TX), α-SHH (Cell Signaling, Danvers, MA), α-SUFU (Cell Signaling), α-PTCH1/2 (Cell Signaling), α-SMO (Sigma-Aldrich), α-BCL-2 (Cell Signaling), α-Caspase 3 (Cell Signaling), α-Caspase 9 (Sell Signaling), α-GAPDH (MilliporeSigma, Burlington, MA) and α-PCNA (Cell Signaling). Inhibition of the Hh-signaling pathway was performed using Vismodegib/GDC0499 (Selleckchem, Houston, TX).

## RNA interference

THP-1 cells ($1.0 \times 10^6$) were transfected with human siRNA (10 nM) using Lipofectamine 3000 (Invitrogen, Waltham, MA). All siRNAs were ON-TARGETplus SMARTpool (Dharmacon, Lafayette, Co). Briefly, specific siRNA (3 µl) and Lipofectamine 3000 reagent (7.5 µl) were added to the Opti-MEM medium (250 µl) (Invitrogen), incubated for 5 min at room temperature, and then added to the cell suspension in a 6-well plate. Scrambled iRNA was used as a control in uninfected and infected samples. Post-transfection (24 h), cells were infected with cell-free *E. chaffeensis* (MOI 100). Cells were harvested at 24 hpi and ehrlichial load determined using qPCR as described previously [8]. All knockdowns were performed with three biological and technical replicates and significance determined using a *t*-test analysis.

## Transfection and immunofluorescent microscopy

HeLa cells were transfected with plasmid using Lipofectamine 2000 (Invitrogen) according to the manufacturer's protocol. Cells ectopically expressing TRP120 were imaged by immunofluorescent microscopy as described previously [9]. Intensity correlation analysis (ICA) and corrected total cell florescence (CTCF) were measured using ImageJ [87,88].

## Co-Immunoprecipitation (Co-IP)

Interactions between TRP120 and PTCH2 were determined by Co-IP using the Pierce Crosslink IP Kit (Thermo Fisher Scientific, Rockford, IL). THP-1 cells (100% confluent) were infected with *E. chaffeensis* at an MOI of 100 for 24 h. The cells were harvested, and Co-IPs were performed with TRP120-I1 and PTCH2 antibodies according to the manufacturer's protocol. Immunoprecipitated eluates and starting input lysates were processed for Western blotting and probed for TRP120 or PTCH2. IP control antibody included IgG purified from normal rabbit serum with Melon Gel IgG Spin Purification Kit (Thermo Fisher Scientific). The Co-IPs were performed in triplicate.

## Surface plasmon resonance (SPR)

SPR was used to investigate binding kinetics of rTRP120 tandem repeat domain (TRP120-TR) and rPTCH2 (loop1; putative ligand binding region) and performed as described previously [3]. Briefly, SMFS-AFM was used to directly extract energetic, thermodynamic and kinetic parameters from force curves describing the TRP120-PTCH2 receptor binding free-energy landscape. rPTCH2 was immobilized on a nickel chip followed by the injection of an analyte solution containing solubilized rTRP120-TR or rTrx (-) to determine the binding affinity on a Biacore T100. Constant injection of rTRP120-TR provided a quantified readout derived from the change in mass on the surface of the chip as rTRP120-TR binds rPTCH2 at a constant rate until reaching equilibrium. The unbinding forces were plotted as a function of the loading rate; from this plot we extracted the dissociation rate (koff) and the energy barrier width xu (nm). The kinetic on-rate (kon) was obtained by varying the dwell time of the TRP120-functionalized tip on cell surfaces, thereby determining binding probability. Using these parameters, we quantified the ΔG of interaction. The results were expressed as the means ± standard deviation (SD) of data obtained from three independent experiments. rTrx fusion protein was used as the negative control since it has no effect on Hh signaling. The binding affinity ($K_D$) was determined for TRP120-PTCH2 interaction by extracting the association rate constant and dissociation rate constant from the sensorgram curve ($K_D = Kd/Ka$).

## Confocal microscopy

*Ehrlichia chaffeensis*-infected (MOI 100) and uninfected THP-1 cells were seeded in T-75 flasks (Corning, Lowell, MA) and collected at 0, 2, 4, 10, 24 and 48 hpi. rTRP120-FL, rTrx, rShh, TRP120-TR-Hh, TRP120-TR (-), TRP120-Hh-SLiM and TRP120-Hh-SLiM-mut peptide-stimulated THP-1 cells were collected at 6 hpt. Experiments performed with at least three biological and technical replicates. THP-1 samples (non-adherent) were washed twice and adhered to glass slides by cytocentrifugation (1000 RPM for 5 min). Uninfected, *E. chaffeensis*-infected, rTRP120-FL-, rTrx-, rShh-, and TRP120-Hh-SLiM and TRP120-Hh-SLiM-mut peptide-treated primary human monocytes were seeded in 12-well plates (Corning) containing a coverslip and incubated for 10 h. After incubation, cells were washed twice with phosphate buffered saline (PBS). THP-1 cells and primary human monocytes were fixed with 4% paraformaldehyde (PFA) for 20 min at room temperature followed by three subsequent washes in PBS. Fixed cells were blocked and permeabilized with 0.3% Triton X-100 in 2% BSA for 30 min and washed. The cells were then incubated subsequently with a mouse monoclonal GLI-1 primary antibody (1:200) and in-house rabbit Dsb serum in blocking buffer (PBS with 2% BSA) for 1 h, with PBS washes (3X) after each treatment. Cells were incubated with Alexa Fluor 488 or Alexa Fluor 568 conjugated secondary antibodies goat α-mouse and goat α-rabbit (Thermo Fisher Scientific) diluted 1:200 in blocking buffer for 30 min and mounted with Pro-Long Gold Antifade with DAPI (4′,6-diamidino-2-phenylindole; Thermo Fisher Scientific). Confocal laser micrographs were obtained with Zeiss LSM 880 laser microscope and analyzed with Zen black and Fiji software. For confocal analysis, randomized areas/slide (n = 10) were used to detect GLI-1 nuclear translocation.

## RNA isolation and cDNA synthesis

Uninfected, *E. chaffeensis*-infected, rTRP120-FL, rShh, TRP120-Hh-SLiM peptide and TRP120-Hh-SLiM-mut peptide-stimulated THP-1 cells were harvested at different time points and data was generated from three biological and technical replicates. *E. chaffeensis* infection was collected at 4, 8, 24 and 48 hpi at MOI 100. THP-1 cells were incubated with 50, 500 ng/mL or 1 μg/mL of peptide or recombinant protein and harvested at 24 hpt. rTrx (-) or uninfected/untreated cells were used as controls for infection, and protein and peptide treatments to determine fold-change. Total RNA was isolated from each sample ($10^6$ cells/sample) using RNeasy Mini kit (Qiagen, Hilden, Germany). On column DNA digestion was performed using the RNase-free DNase kit (Qiagen). The concentration and the purity of RNA were determined using a Nanodrop 100 spectrophotometer (Thermo Fisher). cDNA was synthesized from total RNA (1.0 μg) using iScript cDNA Synthesis Kit (BioRad, Hercules, CA) according to the manufacturer's protocol.

## Human Hh signaling pathway PCR array

The human Hh signaling target PCR array (Qiagen) profiled the expression of 84 key genes responsive to Hh signal transduction, that includes receptors, ligands, and transcription factor/co-factors, and known target genes. PCR arrays were performed according to the PCR array handbook from the manufacturer (Qiagen). Real-time PCR was performed using $RT^2$ Profiler PCR array in combination with $RT^2$ SYBR green master mix (Qiagen) using a Quant-Studio 6 Flex real-time PCR system (Thermo Fisher Scientific). PCR conditions and analysis were conducted as previously described [3]. The red, black and green dots in the volcano plot represent upregulation ($\geq 2$), no change or down-regulation ($\leq$ -2), respectively for a given gene on the array. The horizontal blue lines on the volcano plots determine the level of significance ($p \leq 0.05$).

## Western immunoblot analysis

For Western blots, THP-1 cells were harvested and washed twice with PBS and lysates were prepared using CytoBuster protein extraction reagent (Novagen/EMD, Gibbstown, NJ) supplemented with complete mini EDTA-free protease inhibitor (Roche, Basel, Switzerland), phenylmethene-sulfonylfluoride PMSF (10 mM) (Sigma-Aldrich). The cell lysates were centrifuged at 15,000$g$ for 10 min at 4˚C. The supernatants were collected, and protein concentration was then measured using Pierce BCA Protein Assay Kit (Thermo Fisher Scientific). Equal amounts of protein (15–30 μg/well) were separated by sodium dodecyl sulfate-polyacrylamide gel electrophoresis (SDS-PAGE), transferred to nitrocellulose membrane and immunoblotted with primary antibodies. Horseradish peroxidase-conjugated goat anti-rabbit or goat anti-mouse IgG (H+L) secondary antibodies (Kirkegaard & Perry Laboratories, Gaithersburg, MD) were used and visualized by SuperSignal West Dura chemiluminescent substrate or ECL (Thermo Fisher Scientific). All Western blots were performed with at least three biological and technical replicates and significance determined by $t$-test analysis.

## Mitochondrial membrane potential assay

To confirm *E. chaffeensis*-mediated inhibition of host cell apoptosis, infected (MOI 100) and uninfected THP-1 cells were treated with DMSO (- control) or Etoposide (100 μm) (Selleckchem), an inhibitor of topoisomerase II and inducer of cellular apoptosis, for 24 h and stained with a JC-1 Mitochondrial Membrane Potential Detection Kit using the manufacturers protocol (Biotium, Fremont, CA). JC-1 is a cationic carbocyanine dye that accumulates in mitochondria. The dye forms JC aggregates at higher concentrations (orange; 590±17.5nm) in mitochondria with positive membrane potential in normal cells. Inversely, due to depolarization of the mitochondrial membrane in apoptotic cells, JC-1 remains as a monomer and yields green fluorescence (emission of 530±15nm). To inhibit Hh signaling, THP-1 cells were treated with a SMO-specific inhibitor Vismodegib (200 nM) for 24 h. NucView 488 & MitoView 633 Apoptosis Assay Kit was utilized to examine cellular apoptotic state using the manufacturers protocol (Biotium). A micrograph was used to demonstrate that the Nucview488 dye (a substrate of active caspase 3) is cleaved by caspase 3/7 and produces green fluorescence due to the activation of the cellular apoptotic pathway. Experiments were performed with at least three biological and technical replicates.

## Hh inhibitor infection analysis

THP-1 cells were treated with Vismodegib or DMSO (200 nM) and infected with *E. chaffeensis* (MOI 50) for 24 h. THP-1 cells were harvested for Western blot and Diff-Quik staining (Thermo Fisher Scientific). Ehrlichial load was determined using qPCR as described previously [8]. Cell viability and count were measured with the Cellometer mini (Nexcelom, Lawrence, MA) with preinstalled normal and *E. chaffeensis*-infected THP-1 cell profiles. Cellometer Mini uses bright field imaging and pattern-recognition software to count and define individual live cells and dead cells stained with Trypan Blue. An analysis summary is produced, including a Trypan blue cell count, concentration, diameter and % viability. At least three biological and technical replicates were performed.

## Acknowledgments

We thank Jignesh Patel for assistance in isolating the primary human monocytes. We thank the UTMB Solution Biophysics Laboratory and the Optical Microscopy Core for assistance with confocal microscopy.

## Author Contributions

**Conceptualization:** Caitlan D. Byerly, Shubhajit Mitra, LaNisha L. Patterson, Jere W. McBride.

**Data curation:** Caitlan D. Byerly, Shubhajit Mitra, LaNisha L. Patterson, Nicholas A. Pittner, Thangam S. Velayutham, Veljko Veljkovic.

**Formal analysis:** Caitlan D. Byerly, Shubhajit Mitra, LaNisha L. Patterson, Thangam S. Velayutham, Veljko Veljkovic.

**Funding acquisition:** Jere W. McBride.

**Investigation:** Caitlan D. Byerly, Shubhajit Mitra, Veljko Veljkovic.

**Methodology:** Caitlan D. Byerly, Shubhajit Mitra, LaNisha L. Patterson, Jere W. McBride.

**Project administration:** Jere W. McBride.

**Resources:** Slobodan Paessler.

**Software:** Veljko Veljkovic.

**Supervision:** Jere W. McBride.

**Validation:** Caitlan D. Byerly, Veljko Veljkovic, Jere W. McBride.

**Visualization:** Caitlan D. Byerly, Jere W. McBride.

**Writing – original draft:** Caitlan D. Byerly, Shubhajit Mitra, Jere W. McBride.

**Writing – review & editing:** Caitlan D. Byerly, Jere W. McBride.

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
