## [Decision Letter · Decision Letter 0]

1 Apr 2022

Dear Mrs. Byerly,

Thank you very much for submitting your manuscript "Ehrlichia SLiM ligand mimetic activates Hedgehog signaling to engage a BCL-2 anti-apoptotic cellular program" for consideration at PLOS Pathogens. As with all papers reviewed by the journal, your manuscript was reviewed by members of the editorial board and by several independent reviewers. The reviewers appreciated the novelty and significance; while none of the reviewers argued that additional experiments were necessary, consideration of their comments will strengthen the manuscript. Based on the reviews, we are likely to accept this manuscript for publication, providing that you modify the manuscript according to the review recommendations.

Sincerely,

Stacey D Gilk, Ph.D.

Guest Editor

PLOS Pathogens

Raphael Valdivia

Section Editor

PLOS Pathogens

Kasturi Haldar

Editor-in-Chief

PLOS Pathogens

orcid.org/0000-0001-5065-158X

Michael Malim

Editor-in-Chief

PLOS Pathogens

orcid.org/0000-0002-7699-2064

Reviewer Comments (if any, and for reference):

Reviewer's Responses to Questions

**Part I - Summary**

Reviewer #1: This study describes the identification and characterization of a Hedgehog pathway SLiM mimetic within the secreted Ehrlichia chaffeensis TRP120 effector. Authors identified the Hedgehog pathway SLiM mimetic through BLAST and ISM analysis. The Hedgehog pathway benefits E. chaffeensis replication and survival, as shown through iRNA of various pathway components. Through this knockdown analysis, PTCH2 was identified as being a potentially important target. Through fluorescence microscopy, TRP120 and PTCH2 were shown to colocalize in both THP-1 cells and HeLa cells. Direct protein-protein interactions between TRP120 and PTCH2 were demonstrated with reciprocal immunoprecipitations and SPR. E. chaffeensis infection was shown to induce the Hedgehog pathway through PCR arrays and assaying/imaging nuclear accumulation of the transcription factor GLI-1. Purified and recombinant TRP120 induces the Hedgehog pathway activation and nuclear accumulation of GLI-1 (THP-1 and primary human monocytes). The TRP120-Hh-SLiM was shown to be sufficient for inducing the Hedgehog pathway and blocking the peptide with specific antibodies abolished this induction. Finally, the TRP120-Hh-SLiM is shown to upregulate BCL-2 through the Hedgehog pathway, demonstrating a strategy that inhibits apoptosis and promotes the survival of E. chaffeensis.

This is an excellent and thorough investigation of a TRP120 SLiM mimetic that manipulates the Hedgehog pathway. Multiple approaches were used to confirm the mechanism of action, which is neatly tied back to E. chaffeensis’ virulence strategy and survival. Since this is the first reported eukaryotic Hedgehog pathway SLiM mimetic in a bacterial proteome, the findings are novel and significant for the field of intracellular pathogens. A few suggestions are noted to improve clarity of this manuscript.

Reviewer #2: The manuscript describes a study on Ehrlichia chaffeensis activation of Hedgehog signaling via TRP120. The authors show that TRP120 has sequence and functional similarity to Hedgehog ligands and identify a candidate ligand SLiM. Confirmation of E. chaffeensis activation of Hedgehog signaling is done by siRNA depletion, Co-IP and plasma resonance. Results show that depletion of Hedgehod signaling negatively affects E. chaffeensis reproduction. RTRP120-TR is shown to directly interact with with Patched 2. Infection by E. chaffeensis upregulates Hedgehog signaling and target genes. Additional experiments confirm the interaction of SLIM in nuclear locations of GLI-1 and expansion of Hedgehog signaling. Inhibition of Hedgehog signaling increased pro-apoptosis in infected cells.

Overall, the manuscript is thorough, well-written, includes controls and rigor throughout. The mechanisms described are highly novel and provide significant insights into E. chaffeensis infection strategies. Mechanisms of immune evasion by intracellular pathogens such as E. chaffeensis are key to understanding how infections progress and are maintained in the host. As such, the description of how Hedgehog signaling is activated during infection and contributes to inhibition of apoptosis is highly novel and significant within the field.

Reviewer #3: This is a very interesting study that further defines TRP120 activity during Ehrlichia infection. This versatile protein is shown to trigger hedgehog signaling that impacts infection, gene expression, and apoptosis. The manuscript is logically presented and shows data that will be important for the Ehrlichia field, possibly extending to the intracellular pathogen field at large. I have several comments that, while mostly minor, should strengthen the authors' conclusions.

**Part II – Major Issues: Key Experiments Required for Acceptance**

Reviewer #1: 1. Figure 3A: it is unclear what the lower, left two frames are showing in the E. chaffeensis panel

2. Figures 4A and 7A: For THP-1s, why does there appear to be an even greater concentration of GLI-1 around the outside perimeter of the nucleus when compared to nuclear accumulation levels (but not in primary human monocytes)?

Reviewer #2: The authors show that GLI-1 activation is increased during infection. However, it is not addressed as to why all cells appear to have GLI-1 activation, even cells that do not appear to be infected by E. chaffeensis. This question is based on data presented in Figure 4.

Reviewer #3: No major experiments suggested.

**Part III – Minor Issues: Editorial and Data Presentation Modifications**

Reviewer #1: 1. In general, there are many acronyms that make it somewhat difficult to follow the text. This reviewer suggests spelling out some acronyms to improve readability (e.g. “knockdown” vs. KD, “primary human monocytes” vs. PHM, “Hedgehog” vs Hh, etc.)

2. Line 160: What is dsb? Please spell out this acronym and/or explain.

3. Line 248: typo. GL-1 should be GLI-1.

4. Figure 8A: it looks like there may be some very faint GLI-1 signal from TRP120-TR (-), albeit significantly less than TRP120 or TRP120-Hh-SliM.

5. Line 292 and 316-317: consistent labeling throughout the manuscript is suggested: TRP120 Hh SLiM vs TRP120-Hh-SliM, BCL2 vs BCL-2, Hedgehog vs Hh.

6. Line 340: please explain what JC-1 dye stains.

7. Figure 12 E and F: bar graphs appear to be misaligned at the bottom and/or cut off

Reviewer #2: Minor comments:

Line 128: Define ISM here instead of line 139

Line 160: explain what dsb encodes for and how this correlates to infection status

Figures:

Figure 1: writing within the figure is hard to see. Increase font of "NTD, PR and CTD". NTD and CTD not defined in figure legend. I assume it is N-terminal and C-terminal? Please fix for consistency.

Figure 3, 7, 8, 9, 11: please add scale bar to microscopy images and figure legeds

Figure 4B: The representative image in 4B barely shows on small "dot" corresponding to dsb. This is quite different than the above panel at 10 hours.

Figure 7: Scale bar is mentioned in the figure legends but missing from figure.

Figure 11: It is not clear what MOI was used in these experiments? Based on the images provided in the microscopy insets it appears that there are multiple morulae in each cell. The data provided just states the number of cells that were healthy vs apoptic but how many of those cells in each population were infected? It is also unclear as to how many cells were counted in each condition.

Figure 13: this is a very nice model of how signaling is being altered during infection; however, the key player :E. chaffeensis, is missing in the figure and needs to be added in. How do the authors predict that the bacteria fall in this mechanism?

Reviewer #3: General comments:

1. Fluorescence microscopy results should be quantified throughout to enhance the ability to make clear conclusions.

2. Do not use the words "strong" and "weak" when referring to protein-protein interactions. Refer to affinities instead.

3. Be careful when saying that a protein is activated based on localization in a cell. A downstream readout is critical (and often provided later in the manuscript) to make this claim.

Specific comments:

1. Lines 146-147 - can not make claims about sharing biological functions based solely on in silico analyses.

2. Lines 171-172 - can not make claims about interactions based on siRNA knockdown.

3. Lines 250-252 - I disagree with this conclusion based on the data shown in Fig. 7. Clear activation is not apparent.

4. Lines 524-526 - this study did not directly test whether TRP120 prevents apoptosis via Bcl-2. This statement is premature.

Comments on figures:

1. Figure 3 - PDM should be defined. Colors in panel D should be defined. In panel C, a non-interacting protein control should be included to ensure that PTCH2 is not just "sticky" and binding non-specifically to TRP120.

2. Figure 4 - THP-1 images are unclear as presented. It appears that GLI-1 is actually absent in the nucleus of many cells.

3. Figure 6 - SUFU blot results disagree with Fig. 5B results. This should be explained. In panel A, N and C need to be defined.

4. Figure 7 - Panel A - again the THP-1 results are unclear. GLI-1 appears to be present mainly in the cytoplasm. For comparison, Figure 9 microscopy is much clearer and more convincing regarding nuclear localization of GLI-1. Nuclear fractionation and immunoblots would strengthen claims.

5. Figure 11 - to fully accept that model, TRP120 effects on apoptosis in the presence/absence of Vismodegib must be shown.

PLOS authors have the option to publish the peer review history of their article (what does this mean?). If published, this will include your full peer review and any attached files.

Reviewer #1: No

Reviewer #2: No

Reviewer #3: No

Figure Files:

Data Requirements:

Reproducibility:

References:

---

## [Editor Report · Decision Letter 1]

21 Apr 2022

Dear Mrs. Byerly,

We are pleased to inform you that your manuscript 'Ehrlichia SLiM ligand mimetic activates Hedgehog signaling to engage a BCL-2 anti-apoptotic cellular program' has been provisionally accepted for publication in PLOS Pathogens.

Best regards,

Stacey D Gilk, Ph.D.

Guest Editor

PLOS Pathogens

Raphael Valdivia

Section Editor

PLOS Pathogens

Kasturi Haldar

Editor-in-Chief

PLOS Pathogens

orcid.org/0000-0001-5065-158X

Michael Malim

Editor-in-Chief

PLOS Pathogens

orcid.org/0000-0002-7699-2064
---

## [Editor Report · Acceptance letter]

12 May 2022

Dear Dr. McBride,

We are delighted to inform you that your manuscript, "</i>Ehrlichia</i> SLiM ligand mimetic activates Hedgehog signaling to engage a BCL-2 anti-apoptotic cellular program," has been formally accepted for publication in PLOS Pathogens.

Best regards,

Kasturi Haldar

Editor-in-Chief

PLOS Pathogens

orcid.org/0000-0001-5065-158X

Michael Malim

Editor-in-Chief

PLOS Pathogens

orcid.org/0000-0002-7699-2064